# Therapeutic potential of KLF2-induced exosomal microRNAs in pulmonary hypertension

Hebah A. Sindi[1,2,11✉], Giusy Russomanno [1,11], Sandro Satta[1], Vahitha B. Abdul-Salam[1], Kyeong Beom Jo [1], Basma Qazi-Chaudhry [3], Alexander J. Ainscough[1], Robert Szulcek [4], Harm Jan Bogaard[4], Claire C. Morgan [1], Soni S. Pullamsetti [5,6], Mai M. Alzaydi[1,7], Christopher J. Rhodes [1], Roberto Piva [8], Christina A. Eichstaedt [9,10], Ekkehard Grünig[9], Martin R. Wilkins [1] & Beata Wojciak-Stothard [1✉]

Pulmonary arterial hypertension (PAH) is a severe disorder of lung vasculature that causes right heart failure. Homoeostatic effects of flow-activated transcription factor Krüppel-like factor 2 (KLF2) are compromised in PAH. Here, we show that KLF2-induced exosomal microRNAs, miR-181a-5p and miR-324-5p act together to attenuate pulmonary vascular remodelling and that their actions are mediated by Notch4 and ETS1 and other key regulators of vascular homoeostasis. Expressions of KLF2, miR-181a-5p and miR-324-5p are reduced, while levels of their target genes are elevated in pre-clinical PAH, idiopathic PAH and heritable PAH with missense p.H288Y *KLF2* mutation. Therapeutic supplementation of miR-181a-5p and miR-324-5p reduces proliferative and angiogenic responses in patient-derived cells and attenuates disease progression in PAH mice. This study shows that reduced KLF2 signalling is a common feature of human PAH and highlights the potential therapeutic role of KLF2-regulated exosomal miRNAs in PAH and other diseases associated with vascular remodelling.

[1] National Heart and Lung Institute, Imperial College London, London, UK. [2] University of Jeddah, College of Science, Department of Biology, Jeddah, Saudi Arabia. [3] Department of Physics, King's College London UK, London, UK. [4] Amsterdam UMC, VU University Medical Center, Department of Pulmonary Diseases, Amsterdam Cardiovascular Sciences (ACS), Amsterdam, The Netherlands. [5] Max Planck Institute for Heart and Lung Research, Department of Lung Development and Remodeling, Member of the German Center for Lung Research (DZL), Bad Nauheim, Germany. [6] Department of Internal MedicineUniversities of Giessen and Marburg Lung Center (UGMLC), Member of the DZL, Justus Liebig University, Giessen, Germany. [7] National Center for Biotechnology, King Abdulaziz City for Science and Technology (KACST), Riyadh, Saudi Arabia. [8] Molecular Biotechnology Center, Department of Molecular Biotechnology and Health Sciences, University of Turin, Turin, Italy. [9] Centre for Pulmonary Hypertension, Thoraxclinic, Institute for Human Genetics, University of Heidelberg, Translational Lung Research Center (TLRC), German Center for Lung Research (DZL), Heidelberg, Germany. [10] Laboratory of Molecular Genetic Diagnostics, Institute of Human Genetics, Heidelberg University, Heidelberg, Germany. [11] These authors contributed equally: Hebah A. Sindi, Giusy Russomanno. ✉email: sindi@uj.edu.sa; b.wojciak-stothard@imperial.ac.uk

Pulmonary arterial hypertension (PAH) is a severe lung disorder characterised by progressive vascular remodelling and increased vasoconstriction. Endothelial damage followed by proliferation of vascular endothelial and smooth muscle cells underlie the disease pathology[1] and hypoxia and inflammation are known contributory factors[2]. Formation of angio-obliterative vascular lesions, driven by vascular endothelial growth factor (VEGF), is a hallmark of severe PAH[3].

Recently, a missense mutation in the transcription factor Krüppel-like factor 2 (KLF2) gene was identified in a family with autosomal heritable pulmonary arterial hypertension (HPAH), suggesting that KLF2 signalling may be compromised in the disease[4].

KLF2 is activated by shear stress and plays a key role in the regulation of lung function and development[5]. KLF2-null mice exhibit abnormal blood vessel formation, resulting in embryonic haemorrhage and death. Within the blood vessel wall, KLF2 is exclusively expressed in endothelial cells and promotes vascular homoeostasis, counteracting inflammation, vascular leakage, thrombosis and VEGF-induced angiogenesis. KLF2 also inhibits endothelial cell apoptosis while promoting metabolic quiescence and reducing metabolic dependence on glucose[6]. The identification of a missense mutation in KLF2 gene is significant, as accumulating evidence from pre-clinical models of PAH implicates reduced KLF2 signalling in PAH pathogenesis. Inhibition of KLF2 expression correlates with increased severity of pulmonary hypertension (PH) in apelin knockout mice exposed to hypoxia[7]. KLF2 overexpression improves pulmonary haemodynamics in hypoxic rats[8], but can impair liver function[9,10], so other therapeutic approaches need to be identified.

microRNAs (miRNAs) are small (~22 nucleotide long) non-coding RNAs that negatively regulate gene expression at the posttranscriptional level[11]. Recent studies have shown that miRNAs released by the cells in exosomes, small membrane vesicles of 40–100 nm in diameter, can be taken up and modulate recipient cell responses in the immediate neighbourhood as well as in distant organs and tissues[12,13].

Dysregulation of several miRNAs has been demonstrated in human and animal PAH but the choice of which miRNAs to target constitutes a conceptual challenge[14]. For example, the levels of KLF2-dependent miR-150 in plasma exosomes from PAH patients are reduced and correlate with survival[15]. Pioneering work by Hergenreider et al.[16] demonstrated that exosome-mediated transfer of miRNAs from KLF2-overexpressing endothelial cells to underlying vascular SMCs reduces SMC de-differentiation, thus representing a strategy to combat atherosclerosis[16]. Exosomes have gained special interest as carriers of miRNAs because of their transportability and the ability to convey information within the circulatory system[12]. However, the complexity of the exosomal cargo and low production yield are obstacles for clinical translation[17].

Our approach was to determine whether exosomal miRNAs from KLF2-overexpressing endothelial cells have vasculoprotective effects in PAH. We demonstrate dysregulation of KLF2-induced miRNA signalling in endothelial cells and lung tissues from idiopathic PAH (IPAH) patients and heritable PAH patients with a KLF2 mutation and present evidence of homoeostatic and anti-remodelling effects of KLF2-induced miR-181a-5p and miR-324-5p in vitro and in vivo.

## Results
**Endothelial exosomes mimic homoeostatic effects of KLF2.** In order to study the effects of KLF2-induced exosomes, KLF2 was overexpressed in human pulmonary artery endothelial cells (HPAECs) via adenoviral gene transfer. Recombinant KLF showed

nuclear localisation (Fig. 1a) and the level of KLF2 overexpression (~3-fold increase) corresponded to the expression level induced by physiological shear stress (10 dynes/cm$^2$) in medium-size pulmonary arteries[18].

Exosomes were isolated from conditioned media collected from control (AdGFP) and AdKLF2-GFP-overexpressing (AdKLF2) HPAECs and the purity of the obtained fraction was confirmed by Nanosight LM10 particle tracking and exosomal protein marker analysis (Supplementary Fig. 1a–f). Conditioned media contained predominantly exosome-sized (<100 nm in diameter) particles, with approximately ~$2 \times 10^{10}$ particles in each mL of medium. No significant differences in the total exosome number were found among the groups (Supplementary Fig. 1c, d). Purified exosomes were added to the cultured HPAECs at 1:1 donor-to-recipient cell ratio, i.e. exosomes produced by 10 KLF2-overexpressing cells were added to 10 recipient cells in culture, corresponding to ~$10^5$ particles per cell. Internalisation of fluorescently labelled exosomes was evident after 1 h of incubation (Fig. 1b). Small numbers of adenoviral particles (~100 pfu/mL) detected in the purified concentrated exosomal fractions had no impact on KLF2 expression in exosome-treated cells (Supplementary Fig. 2). Transmission electron microscopy of adenovirus-infected HPAECs revealed that virions localised to cell nuclei and were absent in cytoplasmic vesicular structures (Supplementary Fig. 3), suggesting that the viral contaminants may have originated from damaged cells.

Treatment of HPAECs with KLF2 exosomes attenuated apoptosis and reduced hypoxia- and TNF-α-induced activation of the pro-inflammatory transcription factor NFκB (Fig. 1c, d). KLF2 exosomes also inhibited VEGF-induced proliferation (Fig. 1e, Supplementary Fig. 4) in HPAECs, mimicking to a large extent the effects induced by KLF2 overexpression (Fig. 1f–h). Treatment of cells with control (AdGFP) adenoviruses or control exosomes had no significant effect on cell responses to proliferative or inflammatory stimuli (Supplementary Fig. 5).

The transcription factor KLF4 shows functional overlap with KLF2[19] and its deletion from endothelium exacerbates PH in hypoxic mice[20]. KLF4 silencing did not affect exosome-induced responses or KLF2 expression in cells (Supplementary Fig. 6), suggesting that the observed effects were KLF4 independent.

Exosomes have been administered therapeutically in several disease models. Intravenously injected exosomes can be taken up by the vascular endothelium and underlying vascular tissues[21]. Consistently, we observed localisation of fluorescently labelled endothelial exosomes in the vasculature of different organs in mice 4 h after injection (Supplementary Fig. 7). Of the cargo that exosomes carry, miRNAs have attracted most interest in terms of their functional and therapeutic importance[22]. We therefore carried out an unbiased screen of KLF2-induced exosomal miRNAs with the view of establishing a more viable treatment strategy.

**Selection of potentially therapeutic exosomal miRNAs.** miRNA profiling was carried out with miRCURY LNA™ Universal RT microRNA PCR (Human panel I+II) on exosome fractions from control (AdGFP) HPAECs and KLF2-overexpressing (AdKLF2) HPAECs. Three hundred and thirty miRNAs were detected per sample, with 183 miRNAs shared among all samples and 110 miRNAs differentially expressed using a cut off P value < 0.05. Eighty-six of these miRNAs passed a Benjamini–Hochberg correction (P-adj < 0.05) (Supplementary Table 1). A heat map with unsupervised hierarchical clustering performed on the 86 differentially expressed miRNAs is shown in Fig. 2.

The list of exosomal KLF2 miRNAs was then compared with published lists of miRNAs differentially expressed in human PAH and chronic hypoxia and MCT rat models of PAH[15,23]. Eight

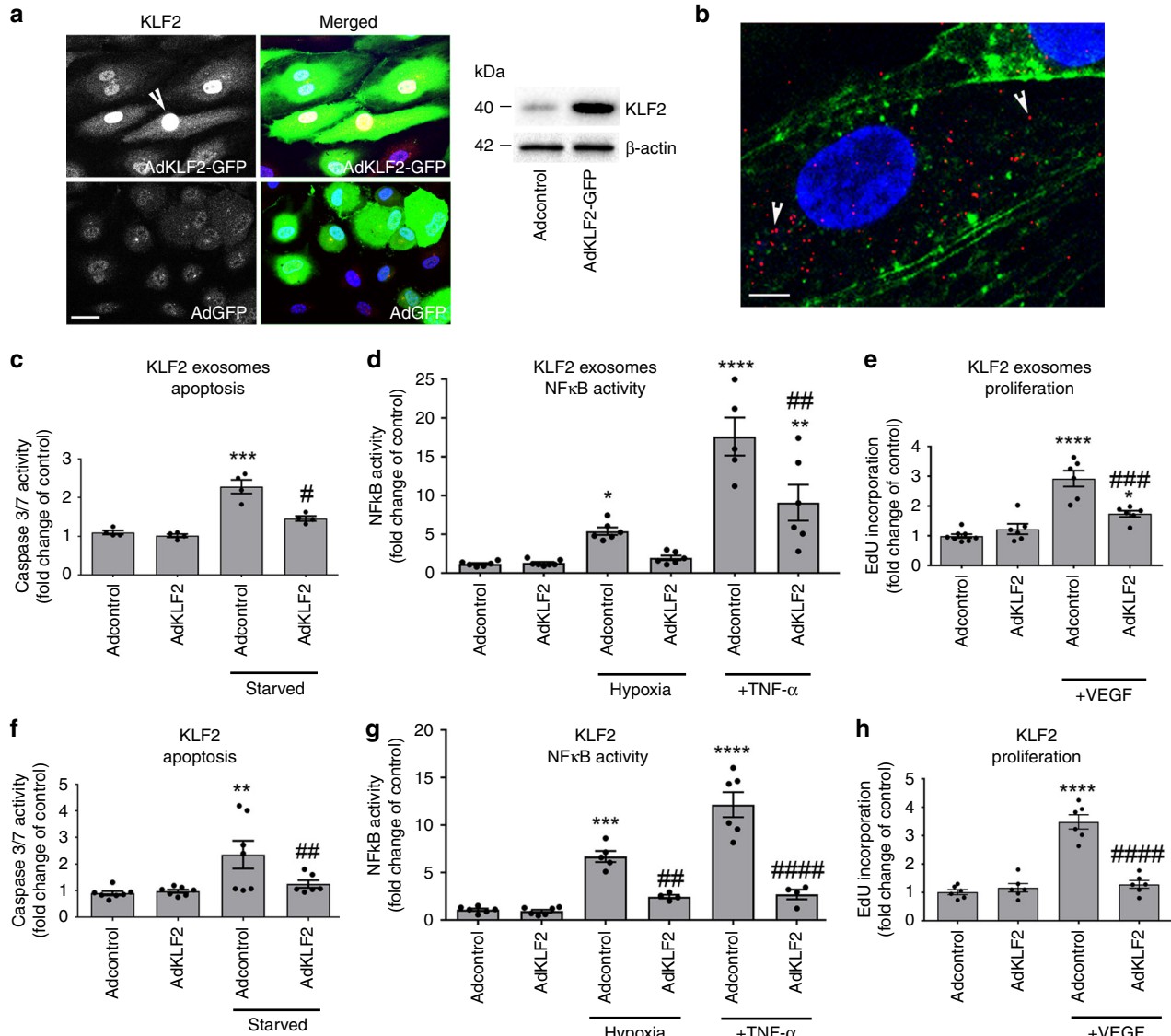

**Fig. 1 Effect of KLF2 and KLF2-induced exosomes on endothelial cell apoptosis, inflammatory activation and proliferation. a** Expression of KLF2 in cells infected with AdGFP and AdKLF2-GFP (24 h). Infected cells are green and the arrowhead points to nuclear localisation of KLF2; bar = 10 μm. **b** Internalisation of PKH26 Red-labelled exosomes (arrowheads) 1 h after treatment. Bar = 2 μm. Effects of **c–e** KLF2-induced exosomes and KLF2 (**f–h**) on caspase 3 per 7 activation in serum-starved HPAECs (24 h), hypoxia- and TNF-α-induced (10 μg/L, 24 h) NFκB activation and VEGF-induced (50 ng/mL, 18 h) cell proliferation in HPAECs, as indicated. Control exosomes were purified from AdGFP-expressing cells, while KLF2 exosomes were purified from AdKLF2-overexpressing HPAECs. **P < 0.01; ***P < 0.001, ****P < 0.0001 comparison with Adcontrol. #P < 0.05, ##P < 0.01, ####P < 0.0001 comparison with treatment controls. One-way ANOVA with Tukey post-test. Values in **c–h** are mean fold-changes of Adcontrols ± s.e.m. In **c** n = 4, **d** n = 6–7, **e** n = 6–8, **f** n = 7, **g** n = 4–6 and **h** n = 6 of independent experiments. Source data are provided in Source Data file.

miRNAs increased by KLF2 but reduced in human or animal PAH were selected for further studies. The selected miRNAs comprised of let-7a-5p, miR-10a-5p, miR-125b-5p, miR-181a-5p, miR-191-5p, miR-30a-3p, miR-30c-5p, and miR-324-5p (Supplementary Table 2).

Exosomes and miRNAs can be transferred between cells at a distance[24]. To verify whether miRNAs can pass from cell to cell under flow conditions, HPAECs were cultured in Ibidi flow chambers connected in tandem (Supplementary Fig. 8). The cells grown in the first chamber were transfected with fluorescent Cy3-miR and then the unbound probe was washed away before the onset of flow typical for proximal pulmonary arterioles[18], at 4–10 dynes/cm². The results show a directional transfer of fluorescent miRNA between endothelial cells under physiological range flow (Supplementary Fig. 8).

**Endothelium-protective effects of miR-181 and miR-324.** To verify the role of the eight selected exosomal miRNAs, HPAECs were transfected with miRNA mimics prior to starvation, hypoxic exposure or stimulation of cells with TNF-α. Transfection efficiency, evaluated with Cy3-labelled miRNA, was ~80 ± 5%.

Only miR-181a-5p and miR-324-5p (referred in the manuscript as miR-181 and miR-324), were protective in all study conditions (Supplementary Fig. 9) and were therefore chosen for further analysis. Interestingly, while single treatments had only partial effect, the combined treatment with miR-181 and miR-324 was significantly more potent (Fig. 3a–d). To see if miR-181 and miR-324 mediate the protective actions of KLF2 exosomes, the exosome-producing cells were transfected with specific miRNA inhibitors. This resulted in a complete inhibition of the endothelium-protective effects of KLF2 exosomes (Supplementary Fig. 10).

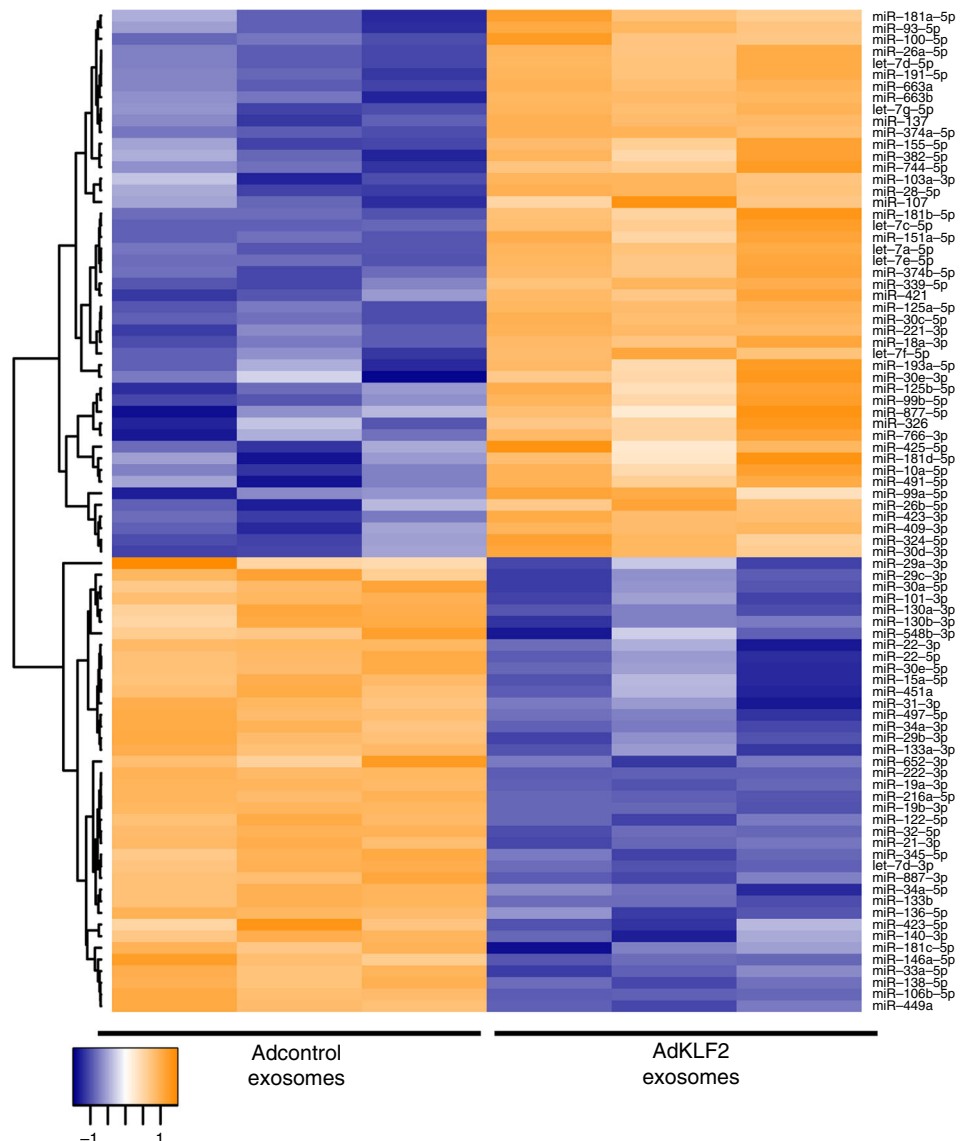

**Fig. 2 KLF2-induced changes in exosomal miRNA expression. a** Heat map and unsupervised hierarchical clustering. The clustering was performed on three samples of exosomal pellets collected 24 h post-infection with AdGFP or AdKLF2-GFP. Each row represents one miRNA, and each column represents one sample. The miRNA clustering tree is shown on the left. The colour scale illustrates the relative expression level of an miRNA across all samples: yellow colour represents an expression level above mean and purple colour represents expression lower than the mean.

miR-181 and miR-324 levels were significantly elevated in KLF2 exosomes and KLF2-overexpressing cells, while the levels of miR-32 were reduced (Supplementary Table 1, Supplementary Fig. 11). Cells treated with KLF2 exosomes showed a significant increase in miR-181 and miR-324 content (Fig. 3e, f). The export ratio (the ratio between the exosomal and the intracellular levels of miRNA) was 1.37 for miR-181a-5p and 1.40 for miR-324-5p, indicating that these miRNAs are actively exported by the cells (Fig. 3g). Expression levels of KLF2, miR-181 and miR-324 were also markedly elevated in flow-stimulated HPAECs (Fig. 3h–j).

To see if miR-181 and miR-324 play a role in endothelial-to-smooth muscle cell communication, human pulmonary artery smooth muscle cells (HPASMCs) were co-cultured with KLF2-overexpressing HPAECs in Transwell dishes. The two cell types were separated by a porous membrane (Fig. 4a), which allows exchange of exosomal particles[16]. HPASMCs co-cultured with KLF2-overexpressing HPAECs showed elevated intracellular levels of miR-181 and miR-324 (Fig. 4b, c), while intracellular levels of

KLF2 remained unchanged (Supplementary Fig. 12a). PASMCs showed reduced hypoxia- and PDGF-induced cell proliferation (Fig. 4d, e), which was prevented by miR-181 and miR-324 inhibitors and the exosome release inhibitor, GW4869[25] (Fig. 4d, e). In contrast to the effects induced by endothelial KLF2 exosomes, direct overexpression of KLF2 did not abrogate PDGF-induced HPASMC proliferation (Supplementary Fig. 12b, c), indicating that KLF2 effects are cell-type specific[26].

**RNA profiling in miR-181 and miR-324-overexpressing HPAECs.** HPAECs transfected with miR-181, miR-324 or non-targeting control miRNA were subjected to RNA profiling to identify targets of potential therapeutic significance. Pairwise differential expression analysis was performed based on a model using the negative binomial distribution and P values were adjusted for multiple test correction using the Benjamini–Hochberg procedure[27].

The list of transcripts that were downregulated by miR-181 and miR-324 with fold-change >1.5 and adjusted P value <0.01 (977

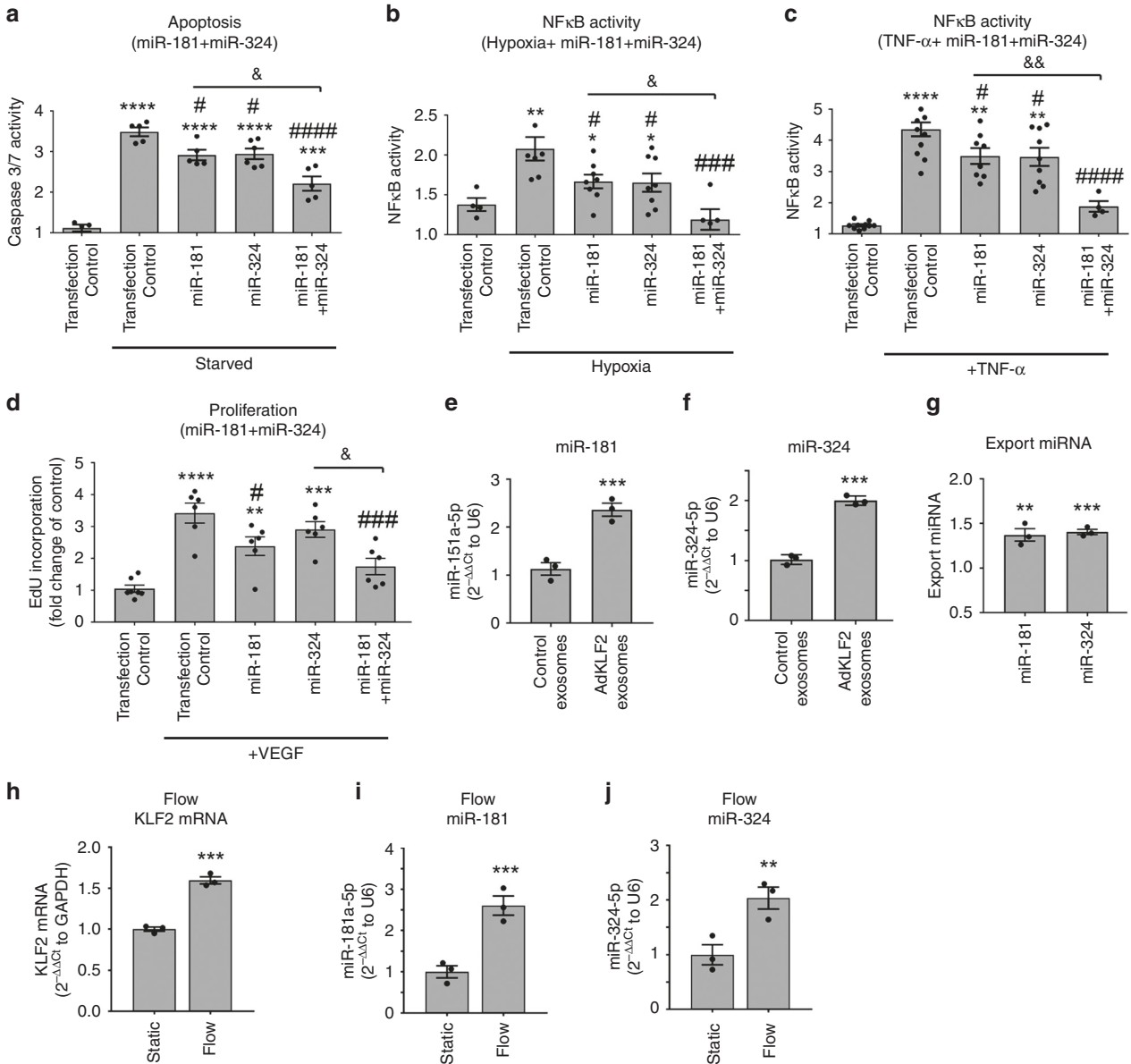

**Fig. 3 miR-181 and miR-324 mediate endothelium-protective effects of KLF2 exosomes. a** Anti-apoptotic, **b**, **c** anti-inflammatory and **d** anti-proliferative effects of single and combined treatments with miR-181 and miR-324. HPAECs transfected with specific miRNAs were serum-starved or were infected with AdNFκB-luc and stimulated with hypoxia or TNF-α (10 μg/L) for 24 h. Alternatively, HPAECc grown in serum-reduced medium were treated with VEGF (50 ng/mL, 18 h). **e** miR-181 and **f** miR-324 levels in HPAECs incubated with control or KLF2 exosomes for 24 h; qPCR. **g** Export ratios for miR-181and miR-324. **h** KLF2 mRNA, **i** miR-181 and **j** miR-324 changes in HPAECs under flow (10 dynes/cm$^2$, 24 h). Bars show mean fold change of untreated controls ± s.e.m. *$P < 0.05$, **$P < 0.01$, ***$P < 0.001$, ****$P < 0.0001$, comparison with transfection controls; #$P < 0.05$, #$P < 0.05$, ###$P < 0.001$, ####$P < 0.0001$, comparison with treatment controls or as indicated; &$P < 0.05$, &&$P < 0.01$, comparisons as indicated. One-way ANOVA with Tukey post-test (**a**–**d**) and unpaired t-test (**e**–**j**). In **a** n = 4–6, **b** n = 4–8, **c** n = 4–12, **d** n = 6–7, **e**–**j** n = 3 of independent experiments. Source data are provided in Source Data file.

for miR-181 and 930 for miR-324) and up-regulated (240 for miR-181 and 251 for miR-324) was compared with the list of predicted in silico target genes. miR-181- and miR-324-mediated targeting of specific mRNAs, including ETS-1 and Notch4 was confirmed by qPCR and luciferase reporter assays (Supplementary Figs. 13 and 14). Thirty-six target genes of miR-181 and 37 target genes of miR-324 were then selected for pathway enrichment analysis (Fig. 5a, Supplementary Fig. 15 and Supplementary Tables 3 and 4). Gene Ontology (GO) and Kyoto Encyclopedia of Genes and Genomes (KEGG) pathway enrichment analysis of targets down-regulated by miR-181 and miR-324 showed significant associations with TNF-α ($P$-adj $= 2.41 \times 10^{-6}$, Fisher's exact test with false discovery rate correction), MAPK ($P$-adj $= 3.6 \times 10^{-4}$),

NFκB ($P$-adj $= 1.1 \times 10^{-4}$), VEGF ($P$-adj $= 1.32 \times 10^{-2}$), and Toll-like receptor signalling pathways ($P$-adj $= 1.2 \times 10^{-4}$; Fig. 5b and Table S5 in the Online Data Supplement). Heat maps and unsupervised hierarchical clustering are shown in Fig. 5c. Key targets of miR-181a-5p associated with inflammation, cell proliferation and vascular remodelling included α-SMA, TNF-α, IL-1, Notch4 and MMP10, while targets of miR-324-5p included MAPK, NFATC2 and ETS-1 (Supplementary Table 5). No significant KEGG pathway associations were found for targets up-regulated by miR-181 and miR-324.

**KLF2, miR-181 and miR-324 and their target genes in PAH.** Blood-derived endothelial colony-forming cells (ECFCs) are often

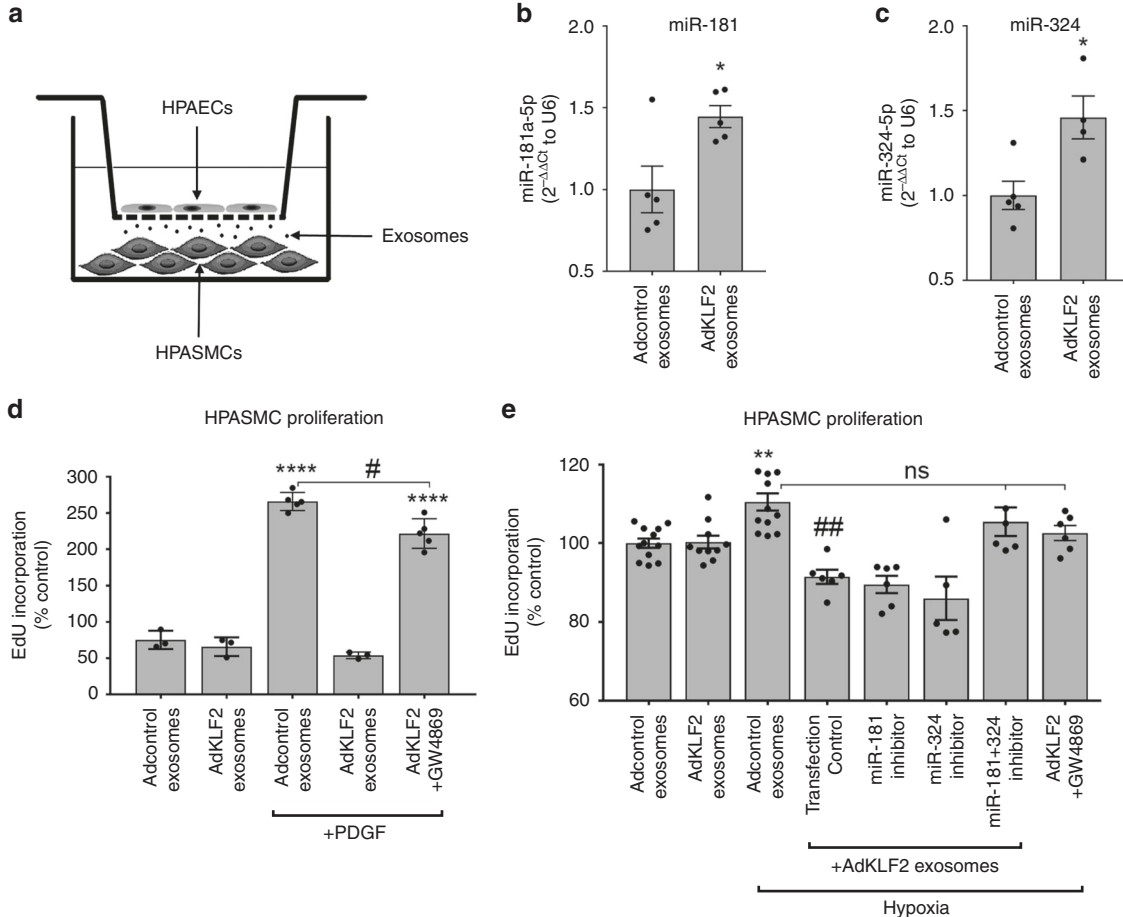

**Fig. 4 The effect of endothelial KLF2 on smooth muscle cell proliferation. a** Transwell dish with KLF2-overexpressing HPAECs separated from HPASMCs by a porous (0.4 μm pore size) membrane. **b, c** miR-181 and miR-324 levels in HPASMCs co-cultured with Adcontrol and AdKLF2-overexpressing HPAECs for 24 h. The effect of endothelial KLF2 exosomes on **d** PDGF (10 ng/mL, 48 h) and **e** hypoxia-induced HPASMC proliferation. HPAECs pre-treated with exosome inhibitor GW4869 (10 μM) were used as a negative control. In **d, e**, bars show the percentage of untreated, untransfected controls; *$P < 0.05$, **$P < 0.01$, ***$P < 0.001$, comparison with Adcontrol or exosomes group; ###$P < 0.001$, comparison with Adcontrol+exosomes, hypoxia group; #$P < 0.05$; &$P < 0.05$, comparisons as indicated. Unpaired $t$-test (**b, c**) or one-way ANOVA with Tukey post-test (**d, e**). Bars are means ± s.e.m. (**b, c**) $n = 5$, **d** $n = 3$–5, **e** $n = 5$–12 of independent experiments. Source data are provided in Source Data file.

used as surrogates for pulmonary endothelial cells in PAH. We examined the levels of KLF2, miR-181 and miR-324 and their selected gene targets in ECFCs from healthy volunteers ($n = 14$) and IPAH patients ($n = 12$). The IPAH cells showed a significant reduction in KLF2 mRNA, miR-181 and miR-324 and a marked increase in the levels of Notch4 (target of miR-181) and ETS-1 (target of miR-324) mRNAs, compared with healthy controls (Fig. 6 a–e).

Notch4 and ETS-1 are key regulators of endothelial proliferation and angiogenesis. Consistent with the elevated levels of Notch4 and ETS-1, IPAH ECFCs showed increased proliferation and tube formation in Matrigel compared with healthy controls (Fig. 6f–h). These responses were markedly reduced in cells transfected with either miR-181 and miR-324 or ETS-1 and Notch4 siRNA (Fig. 6f, h). Reduction in ETS-1 and Notch4 expression in siRNA-transfected cells was confirmed by qPCR (Supplementary Fig. 16).

The RNAscope fluorescent in situ hybridisation, which allows specific identification and quantification of single transcripts[28], showed a marked reduction in the endothelial KLF2 mRNA and increased expression of Notch4 and ETS-1 mRNA in the remodelled vasculature of IPAH patients, compared with healthy lungs ($n = 6$ per group) (Fig. 7a–e). Similar changes in KLF2,

ETS-1 and Notch4 protein expression were noted in lung lysates from IPAH patients, compared with controls (Supplementary Fig. 17).

HPAH patients with c-terminal (p.H288Y) *KLF2* mutation ($n = 3$) also showed a marked upregulation of Notch4 and ETS-1 in the remodelled pulmonary vasculature, identified by endothelial vWF and a prominent α-SMA staining in the small, normally non-muscularised arterioles (Fig. 7f–i). p.H288Y is a heterozygous germline mutation that results in a missense substitution of histidine at amino acid position 288 by arginine in the c-terminal zinc-finger protein domain of KLF2, which has been described as a recurrent somatic mutation in B cell lymphoma[4,29]. Loss of function caused by this mutation in HPAECs was confirmed by reduced nuclear localisation of the p.H288Y (c.862C>T) KLF2 mutant and a loss of anti-proliferative and anti-inflammatory effect in vitro (Supplementary Fig. 18).

**miR-181 and miR-324 supplementation attenuates PH in mice.**
As a proof-of-concept, the Sugen/hypoxia mouse model of PAH was employed to study changes in the expression of KLF2 and miR-181 and miR-324 target genes in the remodelled lung, and to evaluate therapeutic potential of miR-181 and miR-324. In this model, inhibition of VEGF receptor by Sugen (SU5416) induces

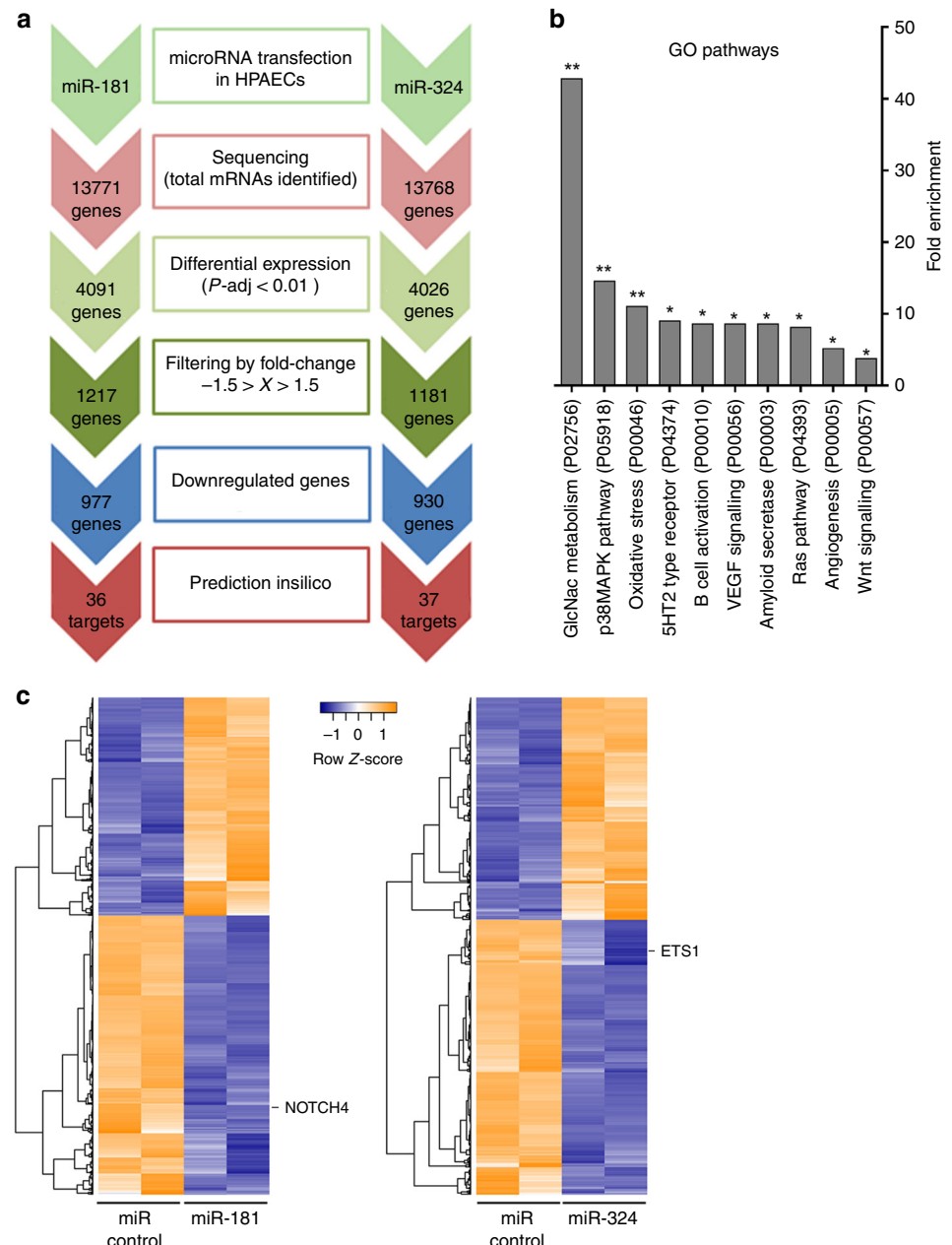

**Fig. 5 Identification of miR-181 and miR-324 targets using RNA sequencing and pathway analysis. a** Flow chart shows identification of potential miRNA targets in PAH. HPAECs were transfected with miRNA mimics and subjected to RNA sequencing. Pairwise differential expression analysis was performed based on a model using the negative binomial distribution and *P* values were adjusted for multiple test correction using the Benjamini–Hochberg procedure; *P < 0.05, **P < 0.01. miRNA target prediction was carried out with TargetScan Human, miRecords and Ingenuity Expert Findings (IPA). **b** Gene enrichment (GO pathways, Panther) in HPAECs transfected with miR-181 and miR-324. **c** Heat map and unsupervised hierarchical clustering. The clustering was performed on two control samples (HPAECs transfected with miRNA control) and two samples of HPAECs transfected either with miR-181 or miR-324. Yellow colour represents an expression level above mean and purple colour represents expression lower than the mean.

an exaggerated vascular repair, driven largely by VEGF signalling, which further increases disease severity[3].

Pilot experiments confirmed the enhanced effect of combined treatment (Supplementary Fig. 19). Three-week exposure to Sugen/hypoxia induced a significant increase in the right ventricular systolic pressure (RVSP), right ventricular hypertrophy (RVH) and pulmonary vascular muscularisation in mice (Fig. 8a–c, h). Intravenous administration of miR-181 and miR-324 resulted in a significant increase in the levels of these miRNAs in the lungs of treated mice (Supplementary Fig. 19). RVSP, RVH and vascular

muscularization were significantly reduced in animals treated with miR-181 and miR-324 (Fig. 8a–h).

The levels of KLF2 mRNA in the lungs of PH mice were significantly lower compared with the lungs of healthy mice (Fig. 8d). Changes in mouse body weight are shown in Supplementary Fig. 20.

A marked reduction in KLF2 mRNA seen in the pulmonary vascular endothelium of Sugen/hypoxia mice was accompanied by increased expression of Notch4 and ETS-1 mRNA (Fig. 8e–j). Conversely, therapeutic supplementation of miR-181 and miR-324

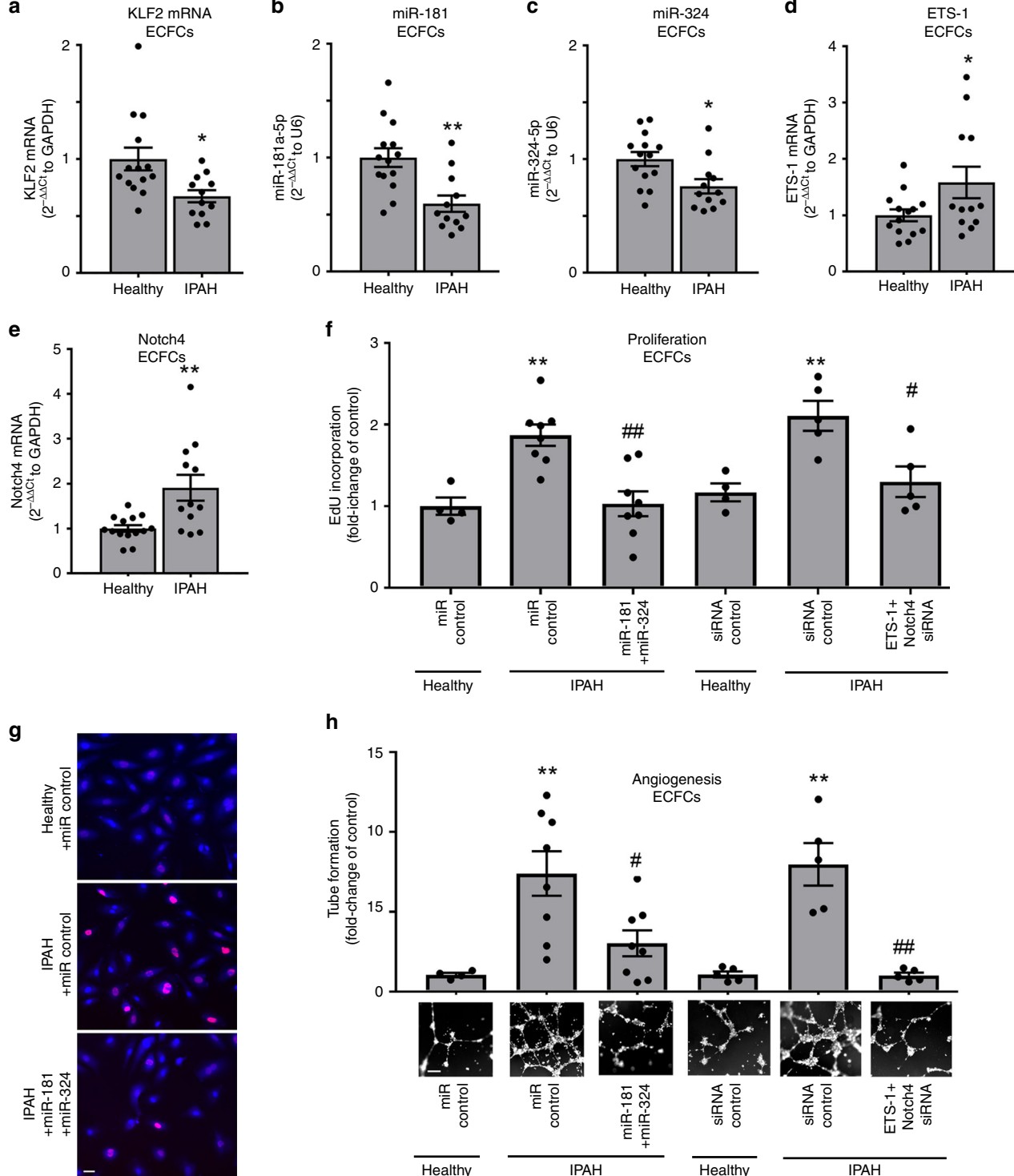

**Fig. 6 Dysregulation of KLF2 signalling in ECFCs from IPAH patients. a** KLF2 mRNA, **b** miR-181 and **c** miR-324, **d** ETS-1 and **e** Notch4 expression in ECFCs from healthy volunteers ($n = 14$) and IPAH patients ($n = 12$). **f, g** Proliferation (EdU incorporation) and **h** angiogenesis (matrigel tube formation) in ECFCs from healthy individuals and IPAH patients, transfected with control miRNA, control siRNA, miR-181 and miR-324 or Notch4 and ETS-1 siRNA, as indicated. Representative corresponding images of tube formation in fluorescently labelled live ECFCs are shown underneath the graph. In **g** EdU-incorporating nuclei are pink. In **g** bar = 20 μm and in **h** bar = 100 μm; *$P < 0.05$, **$P < 0.001$, comparison with corresponding controls; #$P < 0.05$, ##$P < 0.01$, comparison between controls and miR-181 and miR-324- or ETS-1 and Notch4-treated IPAH ECFCs. Bars are means ± s.e.m. In **a–e** $n = 12$–14, unpaired $t$-test and in **f, h** $n = 4$–8, one-way ANOVA with Tukey post-test. All $n$ numbers are biologically independent samples. Source data are provided in Source Data file.

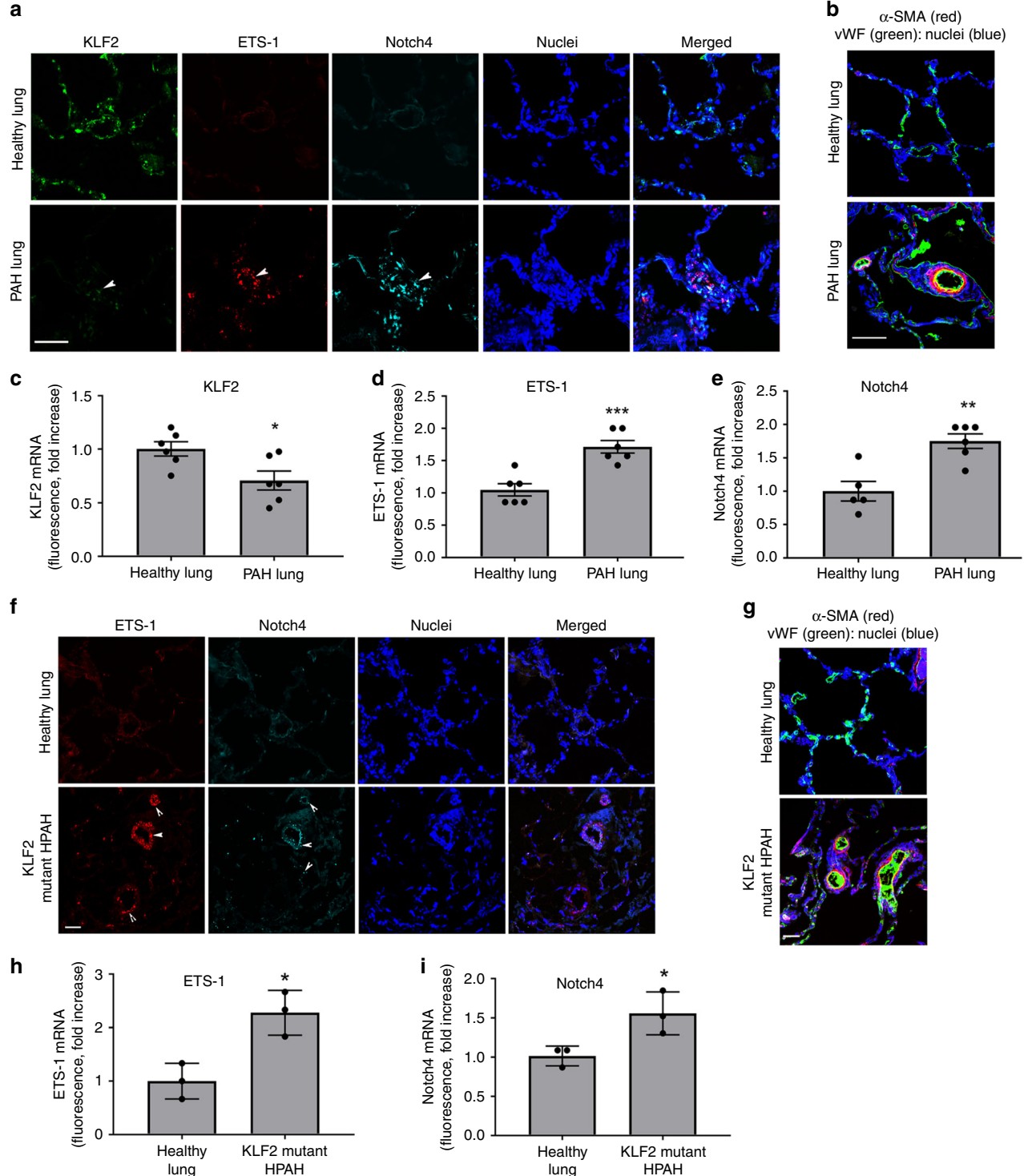

**Fig. 7 KLF2, ETS-1 and Notch4 mRNA levels in lung tissues of PAH patients. a** Representative images of KLF2, ETS-1 and Notch4 mRNA distribution in healthy and PAH lung; RNAscope fluorescent in situ hybridisation. **b** α-SMA and vWF staining in healthy and PAH lung. In **a**, **b**, bar = 50 μm. **c–e** Graphs showing KLF2, ETS-1, and Notch4 mRNA levels in healthy and PAH lung tissues; n = 6. **f** Representative images of ETS-1 and Notch4 mRNA expression in healthy lung and lungs from HPAH patients with disabling *KLF2* mutation. **g** α-SMA and vWF staining in healthy and HPAH lung. In **f**, **g** bar = 25 μm. **h**, **i** graphs showing ETS-1 and Notch4 levels in healthy lung tissues and lung tissues from HPAH patients with *KLF2* mutation, n = 3. In **c–i** values are mean fold-changes of controls ± s.e.m. *P < 0.05, **P < 0.01, ***P < 0.001, comparison with healthy controls; unpaired *t*-test. All *n* numbers are biologically independent samples. Source data are provided in Source Data file.

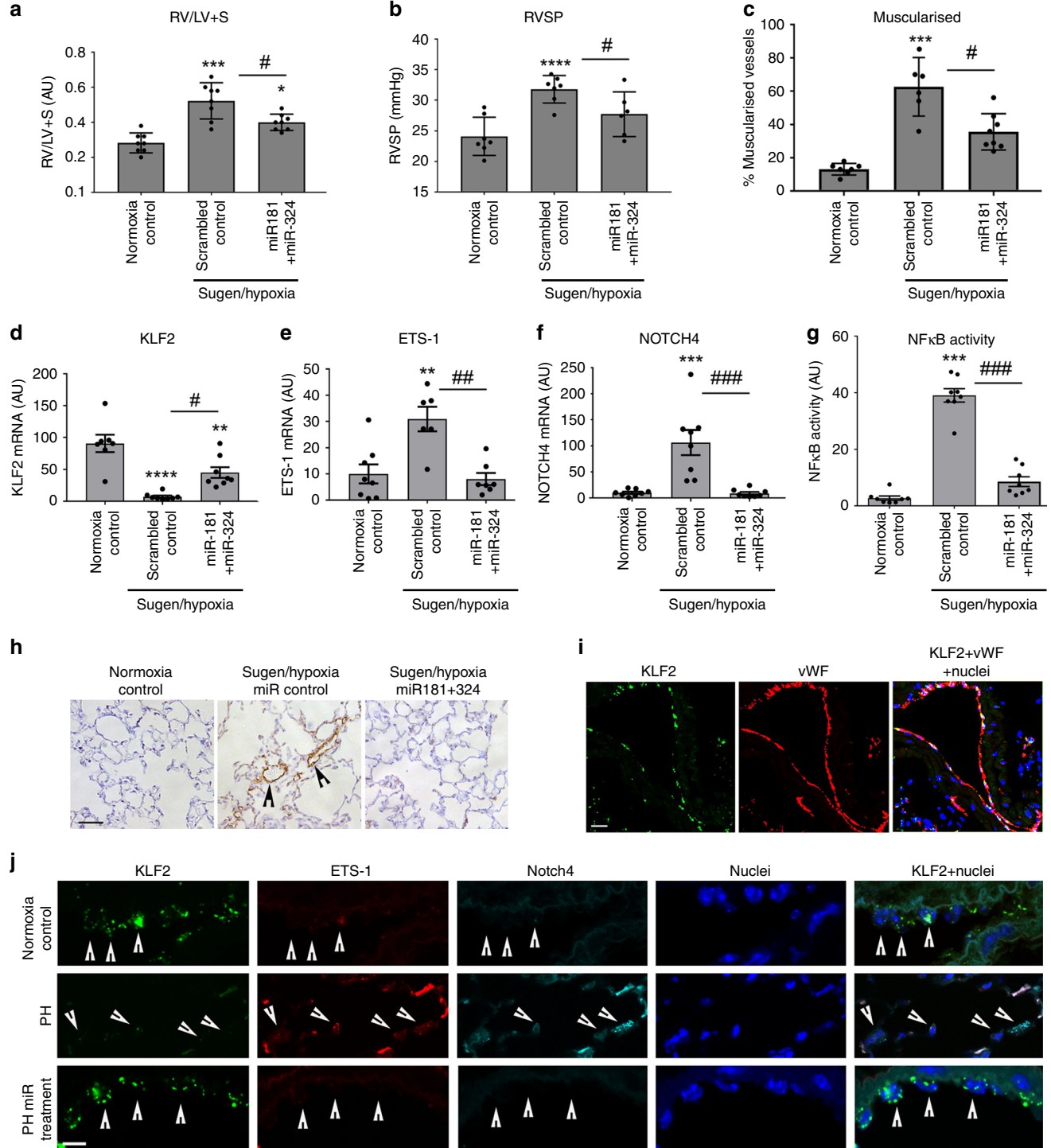

**Fig. 8 Delivery of miR181 and miR-324 attenuates pulmonary hypertension and reduces expression of Notch4, ETS-1 and NFκB activity in Sugen/ hypoxia mice. a** Right ventricular hypertrophy (RV/LV+S), **b** RVSP, and **c** vascular muscularisation in mice treated, as indicated. Changes in **d** KLF2, **e** ETS-1, **f** Notch4 mRNA levels, and **g** NFκB activity (nuclear localisation) in the lungs of Sugen/hypoxia mice; fluorescence (arbitrary units). **\*\***P < 0.01, **\*\*\***P < 0.001, **\*\*\*\***P < 0.0001, comparison with untreated controls; #P < 0.05, ##P < 0.01, ###P < 0.001, comparisons as indicated. Values are means ± s.e.m. One-way ANOVA with Tukey post-test. **h** Representative images of α-SMA staining in the untreated and miR-181+miR-324-treated mice lungs. **i** Co-localisation of KLF2 mRNA with vWF in mouse lung endothelium. **j** KLF2, ETS-1, Notch4 mRNA and nuclei in lung endothelial cells from healthy, pulmonary hypertensive (PH) and miRNA-treated mice, as indicated. Fluorescent in situ hybridisation, confocal microscopy. Each fluorescent dot corresponds to a single transcript; arrowheads point to endothelial cells. In **h**, **i** bar = 25 μm and in **j** bar = 10 μm. In **a** n = 8, **b** n = 6–7, **c** n = 6–8, **d** n = 7–8, **e** n = 6–8, **f** n = 7–8, **g** n = 8 of biologically independent samples. Source data are provided in Source Data file.

resulted in a decrease in the mRNA and protein expression of Notch4 and ETS-1 and α-SMA, reduction in NFκB activity and reduction in the expression of cell proliferation marker, PCNA, in the lung (Fig. 8e–j, Supplementary Figs. 21–24). Interestingly, endothelial KLF2 mRNA expression was partially restored in miRNA-treated mice, possibly as a result of improved flow conditions, secondary to the reduction in vascular remodelling (Fig. 8d–j, Supplementary Fig. 21). Endothelial localisation of the transcripts was confirmed by co-staining with vWF (Fig. 8i).

## Discussion

We are first to provide evidence that KLF2 signalling is a common feature of human PAH and describe therapeutic, cooperative actions of KLF2-induced endothelial miRNAs involving key regulators of vascular homoeostasis, Notch4 and ETS1.

Data from pre-clinical studies and discovery of a missense mutation in KLF2 gene in human HPAH suggest that KLF2 function may be compromised in PAH. While viral delivery of KLF2[30] or treatment with KLF2-induced exosomes[13,16,31] can offset some of the effects of KLF2 inhibition, they have limited applicability in clinical practice[17,32]. We sought to determine whether exosomal miRNAs released by KLF2-overexpressing endothelial cells have a vasculo-protective effect in PAH. We show that delivery of selected KLF2-induced exosomal miRNAs improves pulmonary endothelial cell survival, reduces inflammatory responses and restricts vascular cell proliferation in vitro and in vivo.

To identify mediators of these effects, we conducted an unbiased screen of exosomal miRNAs and selected eight miRNAs of interest, which were upregulated by KLF2 but reduced in PAH, with the view of potential therapeutic intervention. Of these, only miR-181 and miR-324 were protective in all experimental conditions and their effects were enhanced by combined treatment. Functional convergence of multiple miRNAs on disease-relevant pathways has been demonstrated in the prevention of cardiac dysfunction and cancer[14,33–35]. miR-181 and miR-324 targets showed significant associations with key regulatory pathways in PAH, including TNF-α, MAPK, VEGF, NFκB, Toll-like receptor and N-acetylglucosamine (GlcNac) signalling. Numerous functional links among their respective gene targets may explain, at least in part, their synergistic effects observed in vitro and in vivo.

The miR-181 family suppress NF-κB signalling and regulate endothelial cell activation, proliferation and immune cell homoeostasis[36]. Specifically, miR-181a-5p downregulates TNF-α, the key mediator of inflammatory responses in PAH[37]. In vascular SMCs miR-181a inhibits angiotensin II signalling, the adhesion of vascular SMCs to collagen and the expression of α-smooth muscle cell actin, suggesting that miR-181 may negatively modulate proliferative and migratory behaviour of HPASMCs in PAH[38]. We observed contemporaneous, opposing changes in the expression of miR-181 and its target, Notch4, in the remodelled lungs of human PAH patients and Sugen/hypoxia mice. Notch4 predominantly localises to the vascular endothelium and regulates cell apoptosis, proliferation and VEGF-driven angiogenesis[39,40]. Its expression is increased by VEGF and FGF, both abundant in the remodelled PAH lung[3,41]. Expression of Notch 1–4 is elevated in the remodelled lungs of hypoxic rats, with Notch4 mRNA showing the highest (fivefold) increase[42]. Consistent with this, we found that the levels of Notch4 mRNA were significantly elevated in blood-derived IPAH ECFCs, in the plexiform lesions in IPAH lungs[41] and in the remodelled lung tissues from HPAH patients with a KLF2 mutation.

miR-324 controls proliferation, angiogenesis and oxygen metabolism in cells. We noted that miR-324 expression was significantly reduced, while expression of its key target, transcription factor ETS-1, was increased in ECFCs from IPAH patients, lung tissues from IPAH and HPAH patients and lungs from Sugen/hypoxia mice. ETS-1 activation is associated with expansive cell proliferation; high expression levels of ETS-1 facilitate the invasive behaviour of tumour cells and are predictive of the poor prognosis in multiple types of carcinoma[43]. ETS-1 also regulates VEGF-induced transcriptional responses[44], increases endothelial angiogenesis, extracellular matrix remodelling and the expression of glycolytic enzymes and associated feeder pathways[45], known to augment vascular remodelling in PAH[46,47].

miR-181 and miR-324 may also exert their effects by altering post-translational modification of proteins via O-linked (on Ser/Thr) or N-linked attachment (on Asn) of GlcNAc or via attachment of GlcNAc onto fucose residues on the Notch receptors, essential for their activation and ligand specificity[48]. While O-GlcNAcylation is required for vascular adaptation to stress conditions, a chronic response has been linked to vascular injury and increased vasoconstrictor/reduced vasodilator responses in cardiovascular diseases, including arterial hypertension[49,50].

ECFCs from IPAH patients display many of cellular abnormalities associated with the disease, including hyperproliferation, mitochondrial dysfunction and impaired ability to form vascular networks, reflective of the exaggerated repair processes in the remodelled lung. We observed that proliferative and angiogenic responses of IPAH ECFCs were reduced by overexpression of miR-181 and miR-324 and by silencing of Notch4 and ETS-1.

Homoeostatic effects of KLF2 exosomes are likely to result from the cumulative reduction in expression of multiple targets of miR-181 and miR-324. Considering that miRNAs regulate the expression levels of multiple genes simultaneously and often cooperatively[11], delineating individual contributions of miR-181 and miR-324 gene targets is beyond the scope of this investigation. The mechanism through which KLF2 upregulates expression of miR-181 and miR-324 is unknown and may involve interaction of KLF2 with promoter or enhancer regions of target miRNAs, indirect transcriptional regulation by intermediate agents or CpG demethylation factors resulting in hypomethylation of pri-miRNA promoters or post-transcriptional processes[51–53].

KLF2 encourages selective sorting of miRNAs into exosomes[54] and our data indicate that miR-181a-5p and miR-324-5p are actively exported by the cells. Although the mechanism of miRNA sorting is not well understood[12], specific consensus sequence motifs (EXOmotifs) have been implicated[55]. miR-181a-5p has two EXOmotifs that differ from consensus sequence in one nucleotide (GGAC, UGAC), while miR-324-5p shows three motifs, one with perfect match to consensus sequence (CCCU) and two showing alteration in one nucleotide (UCCG, GGAC).

Mechanisms responsible for the reduction in KLF2 expression and signalling in PAH may involve genetic modification[4], dysregulation of BMPR2 function[4] or inflammation[56]. Loss of nuclear localisation as well as anti-inflammatory and anti-proliferative properties of p.H288Y KLF2 mutant suggests that this genetic modification is likely to exacerbate endothelial dysfunction in patients. Inhibition of KLF2 signalling is likely to result in maladaptive responses to flow and endothelial damage[57]. Interestingly, endothelial cells isolated from the subpleural lung microcirculation of patients with PAH at the time of transplantation or necropsy and submitted to ex vivo high shear stress exhibit delayed morphological adaptation, compared with controls[58]. While endothelial KLF2 and endothelium-derived KLF2-induced miRNAs have homoeostatic, anti-proliferative effect on vascular endothelial and smooth muscle cells, direct activation of KLF2 in vascular smooth muscle cells does not restrict cell proliferation, reflective of complex roles of KLF2 in the vasculature[59].

In summary, we show that homoeostatic effects attributed to endothelial KLF2[60] and laminar shear stress[6] can be mimicked by

KLF2-induced endothelial exosomal miRNAs. This study provides evidence of dysregulated KLF2 signalling in PAH and highlights the potential therapeutic role of KLF2-regulated exosomal miRNAs in PH and other diseases associated with endothelial damage, inflammation, proliferation and propensity for vascular cell enlargement.

## Methods

**Endothelial and smooth muscle cell culture**. HPAECs (PromoCell, C-12241) were cultured in endothelial cell growth medium 2 (ECGM2; PromoCell, C-22111) in polystyrene 75 cm$^2$ culture flasks (Sarstedt, Germany) coated with 10 mg/L solution of bovine plasma fibronectin (Sigma F1141) in PBS. Alternatively, the cells were cultured in 96-well cell culture plates (Sarstedt; 10,000 cells per well) or on fibronectin-coated Thermanox® Plastic Coverslips (13 mm; Thermo Scientific, UK, 174950) in a 24-well plate (Sarstedt; 50,000 cells per well, respectively).

HPASMCs (Lonza, CC-2581) were cultured in 75 cm$^2$ tissue culture flasks in smooth muscle cell growth medium 2 (SMCGM2, PromoCell, C-22062), supplemented with 5% FCS and growth factor supplement (PromoCell, C-39267) containing EGF (0.5 ng/mL), Basic FGF (2 ng/mL), insulin (5 µg/mL), and 1% streptomycin and penicillin (100 µg/mL, ThermoFisher, 15070063). Only actively growing cells (~48-h population doubling time), showing morphology typical for endothelial cells (cobblestone when confluent, VE-cadherin$^+$) or smooth muscle cells (elongated, mesenchymal, α-SMA$^+$), were used for experiments.

The cells were cultured under normoxic conditions in a humidified incubator (37 °C, 20% O$_2$, 5% CO$_2$) until 80% confluency and were used between passages 4 and 10. In some experiments, the cells were incubated with human recombinant TNF-α (R&D, 210-TA-020; 10 µg/L), VEGF165 (R&D, 293-VE; 50 ng/mL) or exposed to hypoxia (5% CO$_2$, 2% O$_2$) for 24–72 h.

For non-contact co-culture of HPAECs and HPASMCs, 6.5, 12 and 24 mm Transwell dishes with 0.4 µm pore polyester membrane inserts (Scientific Laboratory Supplies, UK, 3470, 3460, and 3450) were used. The semi-permeable membrane in Transwell insets allows exchange of exosomes between the two cell types[16]. HPAECs were seeded into the fibronectin-coated top chambers inserted into a 24-, 12- or 6-well Transwell dish at 30,000, 100,000 and 200,000 cells per well, respectively and cultured in complete ECGM2 medium, whereas HPASMCs were seeded at the bottom of a 24-, 12- or 6-well plate (50,000, 100,000 and 200,000 cells per well, respectively) and cultured in complete SMCGM2 for 24 h. The two cell types were then washed with PBS, combined together and co-cultured in endothelial cell basal medium supplemented with 10% FBS (Sigma-Aldrich, F7524), and selected components of ECGM2 supplement pack (PromoCell, C-22111): EGF (2.5 ng/L), FGF (10 ng/L), IGF (20 ng/L) with 1% penicillin and streptomycin.

**Patient-derived endothelial cells and lung samples**. All investigations were conducted in accordance with the Declaration of Helsinki. Venous blood samples were obtained with the approval of the Brompton Harefield & NHLI and Hammersmith Hospitals Research Ethics committees and informed written consent from healthy volunteers ($n = 14$) and patients with IPAH ($n = 12$). Participants were identified by number. Demographic and clinical features of healthy subjects and patients are shown in the Table S6. Human endothelial colony forming cells (ECFCs) were derived from peripheral blood samples[61]. Venous blood samples (~50 mL) were collected in EDTA vacutainers, diluted 1:1 with phosphate-buffered saline (PBS) containing 0.2% EDTA and 2% foetal bovine serum (FBS) and layered onto Ficoll Plaque PLUS (GE Healthcare, Amersham, UK) for density gradient centrifugation. Peripheral blood monuclear cells (MNCs) were aspirated, washed in PBS and re-suspended in EGM-2 medium (CC-3156; Lonza Biologics, Slough, UK), supplemented with growth factors (CC-4176, EGM™-2 bullet kit, Lonza), 20% FBS (HyClone, Thermo Scientific, South Logan, UT, USA) and 1% antibiotic and antimycotic solution (Gibco, Invitrogen, Paisley, UK).

Cells were seeded (3–5 × 10$^7$ MNCs in 4 mL of medium per well) in six-well plates coated with type 1 rat tail collagen (BD Biosciences, Bedford, MA, USA). The medium was replaced daily for the first week (removing non-adherent cells), and every 2 days thereafter. Individual colonies were harvested by attaching a cloning cylinder (8 or 10 mm in diameter; Millipore, Watford, UK) with vacuum grease and using warm 0.05% TrypsinEDTA solution (Invitrogen) to remove all the cells within the cylinder. Dislodged cells were re-suspended in fresh EGM-2 medium, seeded (3000–5000 cells per cm$^2$) in 1% gelatin precoated culture vessels and propagated when 70–80% confluent. Distinct colonies of cells, capable of serial propagation for at least eight serial passages and clonal growth, and displaying a stable population doubling time as well as endothelial phenotype, were used between passages 4 to 7. Endothelial cell lineage was confirmed with specific markers, FITC-conjugated Ulex europaeus agglutinin-1 (UEA-1) lectin, mouse monoclonal anti-human CD31 (JC/70 A, Dako, Glostrup, Denmark), anti-human VE-Cadherin (sc-9989, Santa Cruz Biotechnology, Santa Cruz, CA) or rabbit polyclonal anti-human von Willebrand factor (A0082, Dako)[61], with Alexa-488 green-conjugated secondary antibodies (Dako). A three-colour FACSCalibur flow cytometer and CellQuest Pro software (BD Biosciences, Oxford, UK) were used to analyse the stained cells by flow cytometry[61].

Surgical samples of lung tissue were acquired from PAH patients at lung transplantation ($n = 6$). Control tissues ($n = 6$) comprised uninvolved regions of lobectomy specimens from patients undergoing surgery for bronchial carcinoma. Tissues were transported on ice, and samples for protein extraction were snap-frozen between 0.5 and 4.0 h before being stored at −80 °C until further analysis; transport times for both control and PAH samples were similar. Lung lysates were prepared as described in ref. [62]. Briefly, lung samples (~200 mg) were added to an ice-cold solution of 1 mL 100 mM phosphate buffer (pH 7.4), containing 1 mM ethylenediamine tetraacetic acid and 1 mM L-dithiothreitol then homogenised for 1 min using PT-K Polytron® Stand Homogenizer (Kinematica AG, Lucerne, Switzerland). Samples were kept on ice throughout the procedure (no protease inhibitors were added) and frozen at −80 °C in aliquots of 0.1 mL until analysed. Clinical information is shown in Supplementary Tables 6 and 7.

**Adenoviral gene transfer**. HPAECs or HPASMCs (80–90% confluent) were infected with adenoviruses adeno-GFP (Adcontrol; Vector Biolabs, 1060) or adeno-KLF2-GFP (AdKLF2; Vector Biolabs, ADV-213187) at the multiplicity of infection (MOI) 1:100 or 1:250, respectively. Three hours later, media were changed and some cells were treated with TNF-α (R&D, 210-TA-020; 10 µg/L), VEGF165 (R&D, 293-VE; 50 ng/mL) or exposed to hypoxia (5% CO$_2$, 2% O$_2$) for 24 h.

**Immunofluorescence of cultured cells**. To stain KLF2, cells grown on coverslips were fixed with 4% formaldehyde, permeabilised with 0.1% Triton X-100 (Sigma-Aldrich, 9002-93-1) and then incubated with methanol at −20 °C for 5 min. After blocking in 2% bovine serum albumin (BSA) (Sigma-Aldrich, A2058) for 15 min, the cells were incubated with mouse monoclonal anti-KLF-2 antibody (Novus Biologicals, NB100-1051; 1:1000) and then FITC-Goat Anti-Mouse IgG (Jackson ImmunoResearch Inc.,115-095-003; 1:200), and F-actin was visualised with 1 µg/mL TRITC-phalloidin (P1951; Sigma). Following immunostaining, cells were mounted in Vectashield Antifade Mounting Medium containing nuclear stain DAPI (Vector Laboratories, H-1200) and examined under the fluorescent confocal microscope (Leica, TCS SP5, Leica Biosystems, Bretton, Peterborough).

**Exosome purification and quantification**. Exosomes were isolated from conditioned media of HPAECs overexpressing AdGFP or AdKLF2-GFP cultured in ECGM2 medium supplemented with 2% exosome-depleted serum (Thermo Fisher Scientific, A2720801). Exosomes were isolated from 10 mL of conditioned media 24 h post-infection using miRCURY™ Exosome Isolation kit (Exiqon; 300102), according to the manufacturer's protocol (http://www.exiqon.com/ls/Documents/Scientific/exosome-kit-cells-urine-csf-manual.pdf). Briefly, 4 mL of precipitation buffer was added to 10 mL of conditioned culture media and incubated for 60 min on a rolling shaker at 4 °C. After that, the media were spun for 30 min at 3200 × g at 20 °C and the resultant exosome pellet was re-hydrated in 200 µL of re-suspension buffer, sonicated for a few seconds on ice and stored at −20 °C. The NanoSight LM10 Particle Size Analyzer and Particle Counter (Malvern Instruments Ltd) was used to track, analyse and obtain the size distribution and concentration measurement of exosomes in liquid suspension. The presence of exosome membrane markers was confirmed by western blotting and exosome marker microarray analysis.

To check if exosome fractions collected from cells overexpressing AdKLF2-GFP contain live viruses capable of inducing KLF2 expression in recipient cells, serial dilutions of purified exosome fractions (10$^{-1}$–10$^{-7}$) were prepared in Dulbecco's Modified culture medium (Gibco, 1013-021) containing 2% FCS and incubated with 1.5 × 10$^3$ HEK293A cell (ThermoFisher Scientific, R70507) per well in 96-well plates. The infected cells expressing GFP were identified by microscopic examination 48 h later. Plaque formation was scored in 10 wells per dilution following 10-day incubation. The TCID50 (Median Tissue Culture Infectious Dose) was calculated according to the following formula based on the Sperman–Karber method (Hillar Kangro Brian Mahy, Virology Methods Manual 1st Edition; eBook ISBN: 9780080543581; Academic Press 1996):

LogTCID$_{50}$ = highest dilution giving 100% CPE + 0.25−0.5 (total no wells with cpe per 10).

For experiments with HPAECs, the purified exosome fraction was further diluted 50-fold in culture media and 24 h later KLF2 expression in cells was measured by western blotting. Membranes were probed with a rabbit polyclonal KLF-2 antibody (1:1000, Abcam, ab203591) and mouse monoclonal antibody against β-actin (Sigma, A2228, 1:10,000 dilution). Secondary antibodies used were HRP-linked goat anti-rabbit IgG (Sigma-Aldrich, A6154; 1:3000) and HRP-linked sheep anti-mouse IgG (GE Healthcare Life Sciences, NAG31V; 1:2000). KLF2 peptide (US Biological, K1891-37p) was used as a positive control.

**Western blot analysis of exosome marker proteins**. Protein content in exosome fractions was measured with a Pierce BCA Protein Assay Kit (Thermo Scientific). Exosome samples in sample buffer were loaded onto 10% SDS/PAGE gels (10 µg protein per lane) and subjected to electrophoresis followed by western blotting. The membranes were probed for three different exosome markers: cluster differentiation protein 63 (CD36), heat-shock protein 70 kDa (HSP70) and tumour susceptibility gene 101 (TSG101) with primary antibodies: mouse monoclonal anti-CD63 antibody (E-12) (Santa Cruz; sc-365604; 200 µg/mL); mouse monoclonal

anti-HSP 70 antibody (6D444) (Santa Cruz; sc-71278; 200 μg/mL); mouse monoclonal anti-TSG101 clone 51/TSG101 (RUO) antibody (BD Transduction Laboratories; 612696; 250 μg/mL). HRP-linked Goat Anti-Rabbit IgG (Sigma-Aldrich, A6154; 1:3000) and HRP-linked Sheep Anti-Mouse IgG (GE Healthcare Life Sciences, Cat. No. NAG31V; 1:2000) were used as secondary antibodies.

Bands were visualised using Luminata Crescendo Western HRP Substrate (Millipore) in a ChemiDoc™ Imager (Bio-Rad). The relative intensity of the immunoreactive bands was determined by densitometry using Image J software.

**Exosome array analysis**. Purity of exosome fraction was also studied with exosome array analysis (System Biosciences, EXORAY-4) which includes eight different types of antibodies directed against exosome markers: FLOT1, ICAM, ALIX, CD81, CD63, EpCAM, ANXA5 and TSG101, according to the manufacturer's protocol (https://www.systembio.com/downloads/Manual_EXAB__EXORAY_EXEL_WEB.pdf).

**Transmission electron microscopy**. For electron microscopy, adenovirus-infected HPAECs (24 h post-infection) cultured on coverslips were fixed overnight at 4 °C in 2–5% gluteraldehyde in 0.1 m Sörensen's phosphate buffer (pH 7.4), postfixed for 2 h at room temperature in 1% osmium tetroxide and dehydrated with increasing concentration of ethanol before embedding in Araldite II resin. Transversally cut sections were stained with uranyl acetate and lead citrate, and viewed on a transmission electron microscope, Tecnai12 Biotwin with LAB6 filament, using a 2K Eagle CMOS camera. Images were acquired at 120 kV with 100 μm C2 aperture at spot size 6.

**Cell treatment with exosomes and exosome internalisation**. Exosomes were purified from 10 mL of conditioned medium, providing the total of $4.12 \pm 0.13 \times 10^{11}$ particles. Exosomal pellet was re-suspended in ECGM2 and approximately $2 \times 10^5$ exosomes per cell were added to HPAECs grown in 96- or 24-well dishes and incubated for 24 h. To visualise the internalised exosomes, exosomal cell membrane was stained with PKH26 Red Fluorescent Cell Membrane Linker Kit (Sigma-Aldrich; PKH26GL-1KT), prior to the addition of exosomes to the cells, according to the manufacturer's protocol (http://www.sigmaaldrich.com/content/dam/sigma-aldrich/docs/Sigma/Bulletin/pkh26glbul.pdf).

Briefly, following incubation with PKH26, the exosomes were washed three times with PBS by ultracentrifugation at $100,000 \times g$ for 10 min at 4 °C to remove the unbound stain, and were seeded onto endothelial cells cultured on plastic coverslips in 24-well dishes. Following 2 h incubation at 37 °C, the cells were fixed in 4% formaldehyde and permeabilised with 0.1% Triton X-100 in PBS. The cells were then incubated with 1 μg/mL FITC-phalloidin (P5282; Sigma) to visualise filamentous actin, washed 3× in PBS and the coverslips were mounted in Vectashield with DAPI. The cells were visualised under the Leica TCS SP5 Confocal Microscope with the objective ×63.

To visualise exosomal distribution in vivo, single injection of PKH26-labelled exosomes[63] (~$2 \times 10^{11}$ particles in 100 μL of PBS) in healthy mice was used to follow distribution of exosomes in lungs, heart, liver and kidney 4 h post-injection. Lungs, livers and kidneys were collected, fixed, paraffin embedded, cut into sections and stained for von Willebrand Factor (vWF) or H&E. Localisation of the fluorescent mimic to the endothelium was confirmed by co-immunostaining for vWF.

**Universal RT microRNA PCR Human panel I+II**. miRNA profile in HPAECs overexpressing AdGFP (controls) and AdKLF2-GFP was analysed with Exiqon miRCURY LNATM Universal RT (microRNA PCR Human panel I+II). The universal RT microRNA PCR was performed on three exosome fractions per group in two groups: (1) control (AdGFP-overexpressing) HPAECs and (2) HPAECs showing ~3-fold increase in AdKLF2-GFP increase in expression.

All experiments were conducted at Exiqon Services, Denmark. Exosomes were lysed in 700 μL QIAzol Lysis Reagent (Qiagen) in a Qiagen TissueLyzer with one 5 mm stainless steel bead. The lysate was transferred to a new tube with 140 μL chloroform, mixed, incubated for 2 min at room temperature and centrifuged at $12,000 \times g$ for 15 min at 4 °C. The upper aqueous phase was transferred to a new tube and 1.5 volume of 100% ethanol was added. The contents were mixed gently, transferred to a Qiagen RNeasy® Mini spin column, and total RNA was extracted from the exosomes using the miRNeasy® Mini Kit (Qiagen) following the manufacturer's instructions. The RNA was eluted in 50 μL of RNase-free water and stored at −80 °C until analysed.

Nineteen microliters of RNA were reverse transcribed in 95 μL reactions using the miRCURY LNATM Universal RT microRNA PCR, Polyadenylation and cDNA synthesis kit (Exiqon). cDNA was diluted 50× and assayed in 10 μL PCR reactions according to the manufacturer's protocol. Each microRNA was assayed once by qPCR on the microRNA Ready-to-Use PCR, Human panel I+II using ExiLENT SYBR® Green master mix. A "no template" sample in the RT step was included as a negative control and profiled like the samples. The amplification was performed in a LightCycler® 480 Real-Time PCR System (Roche) in 384-well plates. The amplification curves were analysed using the Roche LC software, both for determination of Cq and for melting curve analysis.

To ensure that the quality of the input RNA, the RNA spike-in kit (Exiqon) was used. The RNA isolation controls (UniSp2, UniSp4 and UniSp5) were added during the purification step to detect any differences in extraction efficiency,

whereas the cDNA synthesis control (UniSp6) was added in the reverse transcription reaction. In addition, a DNA spike-in (UniSp3) consisting of a premixed combination of DNA template and primers was add to all panels.

The amplification efficiency was calculated using algorithms similar to the LinReg software. All assays were inspected for distinct melting curves and the Tm was checked to be within known specifications for the assay. To be included in the further analysis, assays had to be detected with 5 Cqs less than the negative control. The upper limit of detection was set to Cq = 37. Cq was calculated as the second derivative. All data were normalised to the average of assays detected in all samples ($n = 9$)[64].

The list of KLF2-induced miRNAs was then compared with the published lists of differentially expressed miRNAs in PAH patients and PH animals[15,23] using Ingenuity Pathway Analysis software (IPA, Qiagen). miRNAs that were reduced in severe PAH but significantly elevated by KLF2 were selected for further analysis.

**Cell transfection**. Transfection of miRNA mimics and inhibitors was carried out with Lipofectamine RNAiMAX Transfection Reagent, according to the manufacturer's instructions (Thermo Fisher, 13778150). To select endothelium-protective exosomal miRNAs of potential therapeutic importance in PAH, HPAECs were left untreated (control) or were transfected with control miRNA (non-targeting transfection control; Ambion Life Technologies, 4464076) or let-7a-5p (ID MC10050), miR-10a-5p (ID MC10787), miR-125b-5p (ID MC10148), miR-181a-5p (ID MC10421), miR-191-5p (ID MC11717), miR-30a-3p (ID MC10611), miR-30c-5p (ID MC11060), miR-324-5p (ID MC10253) at a concentration of 10 nmol/L (all miRVana™ miRNA mimic, 4464066; Ambion Life Technologies). Cy3 dye-labelled Pre-miR Negative Control (10 nmol/L; Ambion, AM17120) was used to assess transfection efficiency.

In following experiments, HPAECs were left untreated or were transfected with control miRNA at 20 nmol/L or control miR (10 nmol/L) plus either miR-181a-5p (10 nmol/L; miRVana™ miRNA mimic, 4464066, ID MC10421; miRVana™ miRNA inhibitor, 4464084, ID MH10421) or miR-324-5p (10 nmol/L; miRVana™ miRNA mimic, 4464066, ID MC10253; miRVana™ miRNA inhibitor, 4464084, ID MH10253) or a combination of miR-181 and miR-324 together (10 nmol/L of each miRNA).

After 5 h, the media were changed and cells were exposed to hypoxia or treated with TNF-α for 24–72 h. Alternatively, on the following day, the untransfected and transfected cells were starved for 9 h before caspase 3/7 assay.

Transfection of cells grown in 24-well dishes with 20 pmol of Silencer Select negative control #1 (Ambion, 4390843), Silencer Select ETS-1 siRNA (Ambion, 432420, ID: S4847) and Silencer Select Notch4 siRNA (Ambion, 4392420, ID: S9643) or siETS-1 and siNotch4 in combination (10 pmol each) diluted in Opti-MEM® I Reduced Serum Medium (Invitrogen, Cat. No. 31985-062) was carried out with Lipofectamine RNAiMAX Transfection Reagent as previously described. The cells were used for experiments 24–48 h post-transfection. Silencing of KLF4 was carried out with Silencer Select KLF4 siRNA (Ambion, 4392420, ID: S17793). Cell proliferation and NFκB activity assays were carried out 48 h post-transfection.

**Lentiviral overexpression of KLF2 c.862C>T p.H288Y mutant**. Lentiviral constructs for KLF2 overexpression were generated as described in ref. [29]. Human KLF2 ORF was generated by PCR from SC127849 cDNA (Origene) using KLF2_EcoRI_FLAG85F CTAGGAATTCGCCACCATGGACTACAAAGACGATGACGACAAG ATGGCGCTGAGTGAACCCAT and KLF2_XhoI_1152R CTAGCTCGAGCTACA TGTGCCGTTTCATGTGC primers, tagged at the N-terminal with flag, and cloned into EcoRI/XhoI sites of pENTR1A no ccdB vector. Mutagenesis of wild-type KLF2 was performed using QuickChange® II Site-Directed Mutagenesis kit (Stratagene), according to the manufacturer's instructions KLF2_C862T_F CAAGAGTTCGTAT CTGAAGGCGCA; KLF2_C862T_R TGCGCCTTCAGATACGAACTCTTG). Wild type and mutant flag-tagged KLF2 were verified by Sanger sequencing and recombined into lentiviral vector pLenti CMV/TO GFP-Zeo DEST (Addgene 17431).

To produce lentiviral stocks, 70% confluent HEK293FT (Invitrogen, cat. R700-07) cells grown in T175 cm² flasks (1 flask per lentiviral preparation), were transfected with packaging plasmids by adding a mix of pMDLg/pRRE (11 μg; Addgene 12251), pRSV-Rev (5.2 μg; Addgene 12253), pMD2.G (7.2 μg; Addgene 12259) and pLenti CMV/TO GFP-Zeo KLF2_C862T (16 μg) dissolved in 1 mL OptiMEM, and 20 mL of PEIpro DNA transfection reagent (VWR, 115-010). Next day, cell medium was replaced with 20 mL of fresh DMEM medium containing 10% FBS supplemented with 0.1 mM MEM non-essential amino acids, 1 mM sodium pyruvate and 2 mM L-glutamine without antibiotics. Lentiviral particles were harvested 48 h later, passed through Steriflip-HV 0.45 μm PVDF filter (Millipore, SE1M003M00). Twenty microliters DNase (1 mg/mL) (1:1000) and 20 μL MgCl₂ (1 M) (1:1000) were added to lentiviral supernatant, mixed by inversion, incubated at 37 °C for 20 min and then incubated overnight at 4 °C with 6 mL of Concentrator-X (631232, Takara). Next day, the mixture was spun at 1500g at 4 °C and the pellet was resuspended in 200 μL, aliquoted and stored at −80 °C.

HPAECs were incubated with lentiviral particles carrying WT KLF2 or KLF2 variant c.862C>T.

To achieve inducible KLF2 expression, HPAECs were incubated with lentiviral particles carrying WT KLF2 or KLF2 variant c.862C>T. KLF2 expression was induced by doxycycline (1 μg/mL) and measured by RT-qPCR at 72 h. Cells were used for experiments 72 h post-transduction.

**Exosomal miRNA transfer under flow**. $6 \times 10^3$ cells in 100 μL ECGM2 medium were seeded into each of six chambers of μ-Slide VI 0.4 (Ibidi, 80606). On the following day, the cells grown in the first chamber were transfected with fluorescent Cy3-miR (10 nmol/L; Ambion, AM17120) and washed several times before connecting the chamber to other two flow chambers in tandem, so that the medium from chamber 1 would flow through chamber 2 and then chamber 3 and out. A peristaltic pump (Ismatec REGLO ICC Digital Peristaltic Pump, Cole-Parmer) created laminar flow of media over the endothelial cells at 4 or 10 dynes/cm². Two independent flow systems were set on one Ibidi slide (two sets of three chambers in two separate flow circuits) and the experiment was repeated in triplicate.

**Caspase 3/7 apoptosis assay**. HPAECs grown in 96-well plates (10,000 cells per well) were left untreated or were treated with exosomes, infected with AdGFP or AdKLF2-GFP or were transfected with miRNA mimics or inhibitors. After an overnight incubation HPAECs were left in full medium or were incubated in serum- and growth factor-depleted medium for 9 h to induce apoptosis.

Apoptosis was measured using Cell Meter™ Caspase 3/7 Activity Apoptosis Assay Kit (AAT Bioquest, ABD-22796). Fluorescence intensity, proportional to the number of apoptotic cells, was analysed in Glomax™ luminometer at Ex/Em = 490/525 nm.

**NFκB luciferase reporter assay**. HPAECs grown in 96-well black polystyrene microplates with clear bottom (Corning, CLS3603) were infected with AdNFκB-luc at the MOI 1:100 for 1 h. The cells were left untreated or were infected with AdGFP (adenoviral control) or AdKLF2-GFP at the MOI 1:250 for 1 h, then the media was changed and the cells were incubated for 24 h in normoxic or hypoxic conditions (5% $CO_2$, 2% $O_2$) at 37 °C. In some experiments, the cells were transfected with non-targeting transfection control or miR-181 or miR-324. TNF-α (10 μg/L) was added to the cells 1 h after adenoviral infection and miRNA transfection and incubated for 24 h. At the end of the experiment, the cells were lysed and luminescence was measured in Promega Luciferase Assay (Promega, E1500) in the Glomax™ luminometer, according to the manufacturer's instructions.

**RNA extraction**. RNA was extracted from trypsinised HPAECs or lung tissue (~10 mg) using the miRCURY™ RNA Isolation Kit (Exiqon), according to the manufacturer's instructions. To remove any residual DNA that may affect downstream applications, an On-Column DNA Removal step was performed using the RNase-free DNase Set (Qiagen). RNA concentration and purity was evaluated using the NanoDrop 2000 spectrophotometer (Thermo Scientific). The $A_{260/230}$ and $A_{260/280}$ ratios were used to assess the presence of contaminants. RNA was then stored at −80 °C for later experiments.

**Real-time quantitative PCR**. Input RNA (50–100 ng/μL) was reverse-transcribed using SuperScrip™ II Reverse Transcriptase (Invitrogen) or TaqMan MicroRNA Reverse Transcription Kit (Thermo Fisher Scientific) and custom Multiplex RT Primer pool in a SimpliAmp™ Thermal Cycler (Applied Biosystems), according to the manufacturer's instructions. The multiplex RT primer pool consisted of primers for miR-181a-5p, miR-324-5p, miR-32-5p and U6 (Thermo Fisher Scientific). No-template samples were included as negative controls.

TaqMan® Gene Expression Assays for KLF2 (Hs00360439_g1, Mm00500486_g1), KLF4 (Hs00358836_m1), NOTCH4 (Hs00965895_g1, Mm00440525_m1), ETS1 (Hs00428293_m1, Mm01175819_m1), GAPDH (Hs02786624_g1, Mm99999915_g1), TNF-α (Hs00174128_m1), MAPK18 (Hs00559623_m1) and TaqMan® miRNA Assays for miR-181a-5p (Assay ID 000480), miR-324-5p (Assay ID 000539), miR-32-5p (Assay ID 002109) and U6 snRNA (Assay ID 001973), all Thermo Fisher Scientific, were used to perform quantitative PCR (qPCR). No-template samples were included as negative controls and all PCRs were performed in triplicate. The reaction was performed on a QuantStudio 12K Flex Real-Time PCR System (Applied Biosystems). Data were analysed using QuantStudio 12K Flex Software version 1.2 (Applied Biosystems). For relative quantification, the data were analysed using the $2^{-\Delta\Delta Ct}$ method, where GAPDH and U6 snRNA were used as endogenous normalisation controls for gene and miRNAs expression, respectively. Additional information regarding TaqMan probes used in this study is provided in Supplementary Table 8.

**Luciferase reporter assays**. Twenty nanograms of each luciferase reporter plasmids harbouring the 3′ UTR of Notch4 (ID HmiT011877; NM_004557.3), ETS-1 (ID HmiT062493; NM_001162422.1) and miRNA Target clone control vector for pEZX-MT06 (negative control; CmiT000001-MT06), all from GeneCopoeia, Inc. (Rockville, MD, USA), were co-transfected with control miRNA, miR-181a-5p or miR-324-5p (Ambion Life Technologies) at 10 nmol/L using Lipofectamine RNAiMAX Transfection Reagent (Thermo Fisher, 13778150) in 70% confluent HPAECs grown in 24-well plates, according to the manufacturer's instructions. Luciferase activities were measured with Luc-Pair Duo-Luciferase HS Assay Kit 2.0 (GeneCopoeia, LF004) in a Glomax™ luminometer (Promega) and were normalised to Renilla luciferase activity.

**Stimulation of KLF2 expression by shear stress**. HPAECs were cultured in 9-cm² Nunc® Lab-Tek™ Flaskettes® (VWR, 62407-340). The slides were detached and inserted into flow chambers in a parallel chamber flow apparatus[65]. Endothelial cells were subjected to laminar flow at 4 or 10 dynes/cm² for 24 h. KLF2 and miRNA expression levels in cells grown under the static and flow conditions were measured by qPCR.

**HPASMC EdU proliferation assay**. Cell proliferation was studied with EdU Cell Proliferation Assay Kit (EdU-594, EMD Millipore Corp, USA, 17-10527), according to the manufacturer's protocol. in total, 30,000 HPAECs were seeded into the Transwell inserts, whereas 50,000 HPASMCs were seeded onto plastic coverslips (Thermo Fisher Scientific, 174950) inserted into 24-well plate. HPAECs were infected with Adcontrol or AdKLF2 and 3 h later the cells were transfected with miRNA inhibitors or transfection controls. Next day, Transwell inserts with HPAECs were placed in the wells containing HPASMCs and the two cell types were co-cultured for further 72 h in normoxic or in hypoxic conditions. Twenty-four hours before the end of the experiment, 10 μmol/L EdU was added to the cells. At the end of the incubation period, the medium was removed, and the cells were fixed in 3.7% formaldehyde solution (Fisher Scientific, F/1501/PB17) in PBS for 15 min at room temperature. After aspiration of the fixing solution, the cells were washed twice with 3% BSA (Sigma-Aldrich, A3294) in PBS (Scientific Laboratory Supplies, LZ51226) and permeabilised with 0.5% Triton X-100 (Sigma-Aldrich, 9002-93-1) for 20 min at room temperature. The cells were then washed twice again with 3% BSA. Two hundred microliters of EdU reaction cocktail prepared according to the manufacturer's instructions were added to the wells. After 30 min incubation in the dark at room temperature, reaction cocktail was removed, and cells were washed three times with 3% BSA.

To visualise cell nuclei, 5 mM stock Nuclear Green DSC1 (Abcam, ab138905) was diluted to 1:1000 in PBS and 200 μl of this solution were added to each well. The cells were incubated for 1 h in the dark at room temperature and then washed two times with PBS. The coverslips were then removed and mounted on a glass slide (VWR, 6310107) in Vectashield® Antifade Mounting Medium (Vector Laboratories, H-1000).

The cells were examined under a fluorescent microscope (Olympus IX70). EdU-incorporating nuclei were visualised at Ex/Em 590/617 nm, while all nuclei stained with Nuclear Green were visualised at Ex/Em 503/526 nm. Cell nuclei were counted using ImageJ software with a macro designed by Steven Rothery (Facility for Imaging by Light Microscopy, Department of Medicine, Imperial College London, UK). Data are shown as fold-change in the percentage of EdU-incorporating cells in hypoxia- or PDGF-treated cells, compared with untreated controls.

To study the effect of endothelial KLF2-induced exosomes on PDGF-induced HPASMC proliferation, HPAECs were seeded in a T-75 flask at 80% confluency and transfected the day after with Adcontrol or AdKLF2, as previously described. After transfection, 8 mL of growth factor- and heparin-free, serum-reduced (1% exosome-depleted FBS) ECGM2 were added to each flask. To verify the role of exosomes in the observed cell responses, HPAECs were pre-incubated with the exosome release inhibitor GW4869 (10 μM, Sigma, D1692)[25] for 8 h, before treatment with adenoviruses. The cells were then incubated in 8 mL of EGGM2 medium in the presence of exosome GW4869. After 24 h, endothelial cell media were collected, centrifuged for 5 min at 10,000g to remove cell debris, and stored at −20 °C.

HPASMCs were seeded onto coverslips inserted into the wells of the 24-well plate (50,000 cells per well) and cultured in normal SMCGM2 for 24 h. The medium was removed, and the cells were starved for 2 h in serum- and growth factor-free SMCGM2. Conditioned media from Adcontrol and AdKLF2-overexpressing endothelial cells were defrosted, supplemented with 10 μM EdU, with or without 10 ng/mL PDGF (Thermo Fisher Scientific, PHG0044) and added to HPASMCs. EdU proliferation assay was performed, as previously described.

**HPAEC and ECFC EdU proliferation assay**. HPAECs (untreated, treated with exosomes, untransfected or transfected with control miRNA, miR-181, miR-324, ETS-1 siRNA, Notch4 siRNA, as appropriate) were grown in 96-well black polystyrene microplates with clear bottom (Corning, CLS3603) at the cell density of 15,000 cells per well. The cells were pre-starved in growth factor-depleted EGM2 medium containing 2% FCS for 8 h. In all, 50 ng/mL of VEGF165 (R&D, 293-VE) was added together with 10 μM EdU 18 h before the end of experiment.

ECFCs from healthy donors and IPAH patients were seeded into 96-well black polystyrene microplates with clear bottom at the cell density of 15,000 cells per well and cultured for 8 h in growth factor-free EGM-2 medium (CC-3156, Lonza Biologics, Slough, UK), supplemented with 5% FBS (HyClone, Thermo Scientific, South Logan, UT, USA) and 1% antibiotic and antimycotic solution (Gibco, Invitrogen, Paisley, UK). Ten micromolar EdU was added to the cells and the cells were incubated for further 18 h. HPAECs and ECFCs were then fixed and stained as described in the EdU proliferation assay for HPASMCs. Instead of Nuclear Green, in some experiments cell nuclei were labelled with 0.1 μg/mL Hoechst 33342 (Thermofisher, 62249) and detected at Ex/Em 361/421 nm.

**RNA-sequencing**. Next-generation RNA sequencing of HPAECs transfected with miR-181 or miR-324 with two biological replicates was performed at the Imperial BRC Genomics Facility (Imperial College London, UK). RNA quality and quantity were assessed using an Agilent 2100 Bioanalyzer (Agilent Technologies) and a Qubit 4 Fluorometer (Thermo Fisher Scientific). RNA libraries were prepared using TruSeq® Stranded mRNA HT Sample Prep Kit (Illumina Inc.) according to the manufacturer's protocol. Briefly, 1 μg of high-quality total RNA (RNA Integrity Number Score ≥8.0) was used for polyadenylated RNA selection using poly-T oligo attached magnetic beads, followed by the fragmentation of poly-A containing mRNA. Cleaved RNA fragments were copied into first strand cDNA using reverse transcriptase with random primers. The cDNA was further converted into double-stranded DNA that was end-repaired to incorporate the specific index adapters for multiplexing, followed by purification and amplification. The amplified libraries were examined using an Agilent 2100 Bioanalyzer and a Qubit. The samples were then pooled and run over four lanes (2 × 100 bp) on a HiSeq 2500 using TruSeq SBS V3-HS kit (Illumina Inc.) in high output run mode. Average sequencing depth across samples was 34.4 million reads.

**Bioinformatics and data analyses**. Sequence data were de-multiplexed using bcl2fastq2 Conversion Software v2.18 (Illumina Inc.) and converted from BCL to FASTQ file format. Base and sequencing quality, GC content, sequence length, duplication, and adapter content were assessed using FastQC (available online at http://www.bioinformatics.babraham.ac.uk/projects/fastqc). Transcripts from paired-end stranded RNA-Seq data were quantified with Salmon v0.8.2 using hg38 reference transcripts[66]. Count data were normalised to accommodate known batch effects and library size using DESeq2[67,68]. Pairwise differential expression analysis was performed based on a model using the negative binomial distribution and P values were adjusted for multiple test correction using the Benjamini–Hochberg procedure[27]. Genes were considered differentially expressed if the adjusted P value was greater than 0.05 and there was at least a 1.5-fold change in expression. Enrichment analysis was performed using DAVID v6.8, Panther and Ingenuity Pathway Analysis (IPA, Qiagen), where significant enrichment was determined if the adjusted P value was >0.05. Network mapping of protein–protein interactions was obtained using STRING v10.5[69].

**Animal experiments**. All animal experiments were conducted according to and in compliance with the guidelines from Directive 2010/63/EU of the European Parliament for animal testing and research, with ethical approval from Imperial College London under UK Home Office Licence number PPL 70/7989 and in compliance with the UK Animals (Scientific Procedures) Act of 1986. All animals were randomly allocated to groups, and all personnel involved in data collection and analysis (haemodynamics and histopathologic measurements) were blinded to the treatment status of each animal. Only weight- and age-matched males were included for experimentation as, in contrast to the human clinical studies, most animal studies have shown that female sex and oestrogen supplementation have a protective effect against PAH.

Eight-week-old C57/BL male mice (20 g; Charles River, UK) were injected subcutaneously with Sugen (SU5416; 20 mg/kg; Tocris Bioscience), suspended in 0.5% [w/v] carboxymethylcellulose sodium, 0.9% [w/v] sodium chloride, 0.4% [v/v] polysorbate 80, 0.9% [v/v] benzyl alcohol in deionized water once per week. Control mice received only vehicle. Mice were either housed in normal air or placed in a normobaric hypoxic chamber (10% O₂) for 3 weeks (n = 8 per group).

miR-181a-5p (ID MC10421) and miR-324-5p (ID MC10253) together, or Negative Control 1 mirVana miRNA mimics In Vivo Ready (all from Ambion), were complexed with Invivofectamine® 3.0 reagent (Invitrogen) and injected intravenously twice a week at 2 mg/kg body weight.

At 3 weeks, the mice were removed from the hypoxic chamber and anaesthetised by intraperitoneal injection of ketamine/dormitor (75 + 1 mg/kg). Development of PH was verified as described previously[70]. RVSP was measured via direct cardiac puncture using a closed-chest technique in the spontaneously breathing anesthetized animal. The animals were then euthanized, the hearts were removed and the individual ventricular chambers were weighed, and RVH was assessed as the right ventricle to left ventricle to septum ratio (RV/LV+S). The right lungs were snap-frozen in liquid nitrogen and stored at −80 °C for biochemical measurements or placed in RNAlater® RNA Stabilisation Solution for RNA isolation. The left lungs were fixed by inflation with 10% formalin, embedded in paraffin, and sectioned for histology. Transverse formalin-fixed lung sections were stained with an anti-smooth muscle actin antibody (Sigma) or Verhoeff's van Gieson stain (EVG) to visualise elastic lamina. Muscularisation of small intrapulmonary arteries was determined by counting all muscularised vessels with a diameter smaller than 50 μm in each section, and expressed as a % of all (muscularised + non-muscularised) vessels. Counting was performed by two blinded investigators.

A preliminary experiment was carried out to evaluate the effects of miR-181 and miR-324 alone or in combination on NFκB activation. Eight-week old C57/BL male mice (20 g; Charles River, UK; n = 5 per group) were injected intravenously with Negative Control 1, miR-181a-5p, miR-324-5p, or miR-181a-5p plus miR-324-5p, as previously described. The day after the mice were injected subcutaneously with Sugen or vehicle and either housed in normal air or placed in a normobaric hypoxic chamber (10% O₂) for 48 h for tissue collection.

**Immunostaining**. For ABC-peroxidase staining, dewaxed and rehydrated lung sections were subjected to heat-induced antigen retrieval in 10 mM sodium citrate (pH 6.0), 0.05% Tween 20, at 80 °C for 10 min, and immunostained using the avidin-biotin-peroxidase complex (ABC Elite, Vector Laboratories) method and 3,3′-dia-minobenzidine as a substrate (DAB Enhanced Liquid Substrate System; Sigma-Aldrich). Sections were incubated with a mouse monoclonal anti-human smooth muscle actin antibody (1:1000; M0851; Dako) and a biotinylated anti-mouse IgG (1:100; BA-2000; Vector Laboratories) as a secondary antibody. Sections incubated with purified non-immune rabbit IgG were used as staining controls. Slides where mounted in DPX mountant for histology (Sigma-Aldrich), and analysed under the under fluorescent confocal microscope (Leica, TCS SP5, Leica Biosystems).

**Tissue homogenate preparation and western blotting analysis**. Tissue samples were homogenised in RIPA buffer (Sigma) for 1 min using PT-K Polytron® Stand Homogenizer (Kinematica AG). Samples were kept on ice throughout the procedure and frozen at −80 °C in aliquots of 0.1 mL until analysed. Protein content was estimated using the bicinchoninic acid method (Thermo Fischer Scientific) with bovine albumin as standard.

Proteins were separated by 4–12% NuPAGE® Bis-Tris gels (Invitrogen) using MES buffer at 200 V constant for 35 min (5 μg equivalent of protein per lane). Membranes were probed with a mouse monoclonal antibody against β-actin (Sigma, A2228, 1:10,000 dilution). Primary antibodies used in western blot analysis included mouse monoclonal anti-β-actin (Sigma A2228, 1:3000), NOTCH4 (Santa Cruz, sc-393893, 1:1000), ETS-1 (Santa Cruz, sc-55581, 1:1000), α-SMA (Sigma, A5228, 1:1000), human/mouse pSer536-p65 (Cell Signaling, 3033, 1:1000) and rabbit polyclonal anti-PCNA (1:500, Santa Cruz Biotechnology, sc-7907). Secondary antibodies were HRP-linked Goat Anti-Rabbit IgG (Sigma-Aldrich, A6154; 1:3000) and HRP-linked Sheep Anti-Mouse IgG (GE Healthcare Life Sciences, NAG31V; 1:2000), rabbit anti-goat polyclonal antibody (R&D, HAF017, 1:3000).

Bands were visualised using Luminata Crescendo Western HRP Substrate) in a ChemiDoc™ Imager. The relative intensity of the immunoreactive bands was determined by densitometry using Image J software.

**Nuclear translocation of p65NFκB**. Nuclear translocation of p65 NFκB was studied with Image J by measuring colocalization of nuclear stain DAPI with p65NFκB stained with rabbit p-NFκB p65 (Ser 276) (Santa Cruz Biotechnology, sc-101749; 1:200) and secondary antibody, TRITC-labelled goat anti-rabbit antibody (Jackson ImmunoResearch Laboratories, 111-025-144, 1:200) in confocal images of cells or lung sections. The white pixel area, marking nuclear NFκB, was used to quantitate p65 NFκB translocation in cells and tissues.

**RNAscope® in situ hybridisation**. Histological sections of lung tissues from treatment-naïve PAH patients at lung transplantation (n = 6), and control tissues comprising uninvolved regions of lobectomy specimens from four patients undergoing surgery for bronchial carcinoma and two unused donor lungs were from the tissue archives at Hammersmith Hospital, Imperial College London. Tissue samples were fixed in 10% formal-saline and embedded in wax, and sections were processed for immunohistochemistry. Sections from HPAH patients with c-terminal missense mutation in KLF2 gene were obtained from three family members, father who died aged 32 and his two daughters who underwent lung transplantation[4]. Lung sections from healthy and Sugen/hypoxia mice were also used for the analysis.

For formalin-fixed, paraffin-embedded tissues FFPE tissues, RNAscope® Multiplex Fluorescent Reagent Kit v2 (Advanced Cell Diagnostics) and TSA™ Cyanine 3 & 5, TMR, Fluorescein Evaluation Kit System (PerkinElmer) were used according to the manufacturers' protocols[28].

Briefly, tissue sections of 5-μm thickness were baked in a dry oven (Agilent G2545A Hybridisation Oven, Agilent Technologies) for 1 h at 60 °C, and deparaffinised in xylene, followed by dehydration in 100% ethanol. Tissue sections were then incubated with RNAscope® Hydrogen Peroxide for 10 min at room temperature. After washing twice with distilled water, manual target retrieval was performed boiling the sections (100 to 103 °C) in 1× Target Retrieval Reagents using a hot plate for 15 min. Slides were then rinsed in deionized water, 100% ethanol, and incubated with RNAscope® Protease Plus at 40 °C for 30 min in a HybEZ hybridisation oven (Advanced Cell Diagnostics). Hybridisation with target probes (human: Hs-ETS1-C1, NM_001143820.1; Hs-NOTCH4-C2, NM_004557.3; Hs-KLF2-C3, NM_016270.2; mouse: Mm-ETS1-C1, NM_011808.2; Mm-NOTCH4-C2, NM_010929.2; Mm-KLF2-C3, NM_008452.2) was carried out incubating the slides at 40 °C for 2 h. Up to three different probes (C1–C3) were multiplexed. After washing twice with Wash Buffer, slides were stored overnight in 5× SSC buffer (0.75 M NaCl, 0.075 M sodium citrate). The following day, the slides were incubated at 40 °C with the following reagents: Amplifier 1 (30 min), Amplifier 2 (30 min), Amplifier 3 (15 min); HRP-C1 (15 min), TSA® Plus fluorophore for channel 1 (fluorescein, cyanine 3, or cyanine 5, PerkinElmer; 1:1000; 30 min), HRP blocker (15 min); HRP-C2 (15 min), TSA® Plus fluorophore for channel 2 (30 min), HRP blocker (15 min); HRP-C3 (15 min), TSA® Plus fluorophore for channel 3 (30 min), HRP blocker (15 min). After each hybridisation step, slides were washed three times with Wash Buffer at room temperature.

In some experiments, RNAscope hybridisation was combined with immunofluorescence. Tissue was blocked for 1 h at room temperature with 3% normal horse serum (Vector Laboratories) in 1× PBS containing 0.1% bovine serum albumin (Sigma-Aldrich), and 0.01% sodium azide (Sigma-Aldrich), and then incubated with polyclonal rabbit antibody raised against human von Willebrand Factor (1:500; A0082; Dako), monoclonal mouse anti-human smooth muscle actin antibody (1:1000; Dako, M0851), or monoclonal rabbit anti-NFkB-p65 (1:400; Cell Signaling Technology, D14E12) at 4 °C overnight. After three washes in PBS, slides were incubated with fluorescently labelled FITC Goat Anti-Rabbit (1:100; 111-095-003; Jackson ImmunoResearch Inc.), TRITC Goat Anti-Rabbit (1:100; 111-025-003; Jackson ImmunoResearch Inc.), and TRITC Goat Anti-Mouse IgG (1:100; T5393; Sigma-Aldrich) for 30 min at RT. Following immunostaining, tissues were mounted in Vectashield with DAPI and examined under a fluorescent confocal microscope (Leica, TCS SP5, Leica Biosystems, Bretton, Peterborough).

**Angiogenesis assay.** Angiogenic responses were studied using matrigel tube formation assay in ECFCs transfected with control (non-targeting) miRNA, miR-181 +324, control siRNA or Notch4 and ETS-1 siRNA. Briefly, 24 h post-transfection the cells were trypsinized, re-suspended in growth factor-free media containing 5% FBS and seeded at a density of 10,000 cells per well in 96-well plates coated with 30 μL Matrigel (BD Biosciences, 354230). Cells were fluorescently labelled with Cell Tracker Green (ThermoFisher Scientific, C2925) following the manufacturer's protocol and examined under a fluorescent microscope Olympus IX70 with ×10 objective. Total tube length was determined using Image J software.

**Statistics.** All experiments were performed at least in triplicate and measurements were taken from distinct samples. Data are presented as means ± s.e.m. Normality of data distribution was assessed with Shapiro–Wilk test in GraphPad Prism 7.03. Comparisons between two groups were made with Student's $t$-test, whereas ≥3 groups were compared by use of ANOVA with Tukey's post hoc test. Statistical significance was accepted at $P < 0.05$.

## Data availability

The RNA-sequencing data generated and analysed during this study are available in the BioProject repository (BioProject ID PRJNA531359; BioSamples SAMN11357006, SAMN11357007, SAMN11357008, SAMN11357009, SAMN11357010, SAMN11357011; Accession numbers SRX5651141, SRX5651140, SRX5651139, SRX5651138, SRX5651143, SRX5651142) at the following link: [https://www.ncbi.nlm.nih.gov/sra?term=SRP191343].

All data generated or analysed during this study are included in this published article (and its supplementary information files) or are available from the corresponding author upon request.

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

## Acknowledgements

This research was supported by Ph.D. Studentships from the Government of Saudi Arabia (Hebah Sindi) and British Heart Foundation project grant PG/16/4/31849. We thank the staff of the Imperial NIHR/Imperial Clinical Research Facility, Hammersmith Hospital (London UK), and Dr. John Wharton for their help in acquiring cells and lung sections from IPAH patients. We also thank Dr. Ines Cebola (Imperial College London) for help with generation of lentiviruses, Dr. Trish Lovell and Dr. Paul Simpson (Imperial College London) for preparation of TEM samples. We acknowledge the support from the Dutch CardioVascular Alliance (DCVA) [2012-08, 2014-11] awarded to the Phaedra and the Reconnect consortium as well as the Impulse grant 2018 awarded to the Phaedra IMPACT consortium. These grants include collective funding by the Dutch Heart Foundation, Dutch Federation of University Medical Centers, The Netherlands Organization for Health Research and Development, and the Royal Netherlands Academy of Sciences. We also acknowledge the support from Associazione Italiana Ricerca sul Cancro (AIRC).

## Author contributions

H.A.S., M.M.A., A.J.A., and S.S. performed in vitro experiments and analysed data; K.B.J., G.R. and V.B.A.-S. performed in vitro and in vivo experiments, immunohistochemistry and analysed data; B.Q.-C. performed exosome quantification; R.S. and H.J.B. provided ECFCs and critically analysed the manuscript; C.C.M. analysed RNA-Seq data; S.P., C.A.E. and E.G. provided HPAH materials and analysis; R.P. provided lentiviral constructs for KLF2 mutatnts; C.R. and M.R. provided IPAH patient microRNA data; B.W.S. secured funding, performed experiments and wrote the manuscript.

## Competing interests

The authors declare no competing interests
