## [Peer Review File · Nature Communications]

Reviewers' comments:

Reviewer #1 (Remarks to the Author):

In this paper by Sindi et al, the authors show that flow-stimulated KLF2 produces exosomes that seem to be protective, and attribute this to the presence of miR-181 and miR-324 regulation of Notch and Ets. The study is very well-performed and links this to the clinical situation by studying cells derived from IPAH patients. A few minor points:

1. The authors never show a direct interaction between the miRs delivered and the the Ets/Notch mRNA. Perhaps a luciferase assay will confirm that which miR is targeting which gene specifically.
2. The transfer of exosomes is thought to mediate these effects, but could the authors show that it is purely the exosomes by inhibiting exosome transfer (GW compound)?
3. While the authors show that miRs are indeed transferred, are the specific miRs actually increased in the recipient cells? Numerous studies show that individual miR levels are quite low and do not necessarily transfer efficiently.
4. The authors show a strong effect on GlcNAc signalling but this does not seem to be investigated further. Are there specific proteins that are affected post-translationally by interruption of GlcNAc signaling?
5. Many journals now require publication of full blots, at least in the supplement. The authors should comply with this growing trend.

Reviewer #2 (Remarks to the Author):

In this submitted manuscript, Sandi et al. profiled the exosomal miRs released from KLF2-overexpressed human pulmonary endothelial cells. In reference to the down-regulated miRs in human PAH and rodent PH models, the authors identified that miR-181a-5p and miR-324-5p are endothelium protector in the context of PAH. They also explored the endothelial-endothelial and

endothelial-smooth muscle cells communications via KLF2 induced exosomes. Further, ETS-1 and NOTCH4 were suggested to be the downstream targets of miR-181 and 324, respectively. These key miRNA and molecules were studied in the human PAH lung specimens. MiR-181 and miR-324 supplements in the sugen+hypoxia PH mice model mitigated PH severity. Overall, the study is nicely executed and the conclusion is convincing. Addressing the following points could further strengthen the manuscript.

1. The involvement of KLF2, miR-181, ETS-1 and Notch4 in PAH has been well documented. The authors need to discuss the novelty of the deduced molecular axis.
2. The authors showed that KLF2-associated exosomes mimics the effects of KLF2 overexpression and shear stress in ECs. I am wondering, whether the KLF2 increased in those exosomes treated ECs? In the recipient cells, is the effects of exosomes KLF4 dependent or independent? The authors should verify whether the KLF2 mRNA or even KLF2 adenovirus vesicles were contained in those exosomes.
3. With KLF2 overexpression in EC, the amount of miR-181 and miR-324 is higher in exosomes than that in the donor cells. Also, the export ratio is higher than 1. These results indicate that KLF2 increases the secretion of these two miRs in exosomes. It is important to address the mechanism by which KLF2 enhances miR-181 and -324 expression and secretion.
4. KLFs may have distinct effect in ECs and SMCs. The authors need to determine whether the KLF2-associated exosomes from EC induces KLF2 in SMCs? What're the functional outcomes if they directly overexpress KLF2 in SMCs?
5. With in silico analysis predict that ETS-1 and NOTCH4 mRNA are direct targets of miR-181 and miR-324, biological validation of this prediction is absolutely necessary. Moreover, authors need to show whether these miRs and their targeting sequences are conserved between human and mouse.
6. The critical molecules in their proposed pathway were verified in the human PAH lung specimens, which is excellent. To further strengthen these results, FISH experiment is suggested to detect miR-181 and miR-342 by in patients' specimens.
7. Since the majority of works are carried out on exosomes, is it possible to deliver human ECs released KLF4-associated exosomes to PH mice? It will be a more direct evidence to show the exosomal related therapeutic effect.

8. Since low flow is associated with PAH (ref. 20). Additional control in Fig. 3h-j should include low flow experiment. A physiological level of flow, e.g., 10 dyne/cm² should also be used in experiments in Fig. S3.

9. Heatmaps generated from RNA-seq should be included in Fig. 5. The protein levels of KLF2, ETS-1, and Notch4 in the lung tissues from PAH patients are necessary to be shown. The authors need to identify cell types shown in Figure 8j?

10. KLF2 overexpression increases exosomal miR-181 and miR-324. However, the total exosome number were not changed. The authors need to use an additional control, namely, a KLF2-decreased miRNA (chosen from the heatmap in Fig. 2) to show the specificity of KLF2 in miR-181 and miR-324 biogenesis, but not exosome formation and secretion.

11. If authors secure KLF2 mutant ECFCs from PAH patients, then they should overexpress miR-181a-5p and miR-324-5p to determine whether the PH phenotype of KLF2 mutant ECFCs can be altered.

Reviewer #3 (Remarks to the Author):

This manuscript addresses the mechanisms that might be relevant to KLF2 in PAH. This gene was recently associated. The authors assess KLF-2 using an adenovirus to overexpress the protein then purify exosomes from HPAEC and control cells and then perform a series of studies to look at function of the control and KLF2 overexpressing exosomes on PAEC function, functions that are relevant to PAH. They then assess miRNAs in those exosomes by microRNA array and discover 100 or so that are changed. After a series of steps then bring this down to 8, then 2 (181a-5p and 324-5p) and perform more studies in vitro and with miRNA supplementation in vivo to assess effects and effects on a couple of key target genes. The manuscript has a number of novel angles, yet the following are relevant comments.

1. There is a disconnect between the beginning, the middle and then the end. It starts with KLF2 genetics (and then that is forgotten), has exosomes and the effect of KLF2 overexpressing exosomes (and then forgotten) and then miR supplementation in the absence of being excapsulated in exosomes. This is limitation to the story telling, as they become different questions. This needs to be addressed. For example, exosomes in the in vivo supplementation, of induce the KLF mutation seen in the genetic form and assess the relevance to the effect. Something linking all this is lost across the paper.

2. There is a distinct lack of inhibition studies. Why not take out the miRNA from KLF-2 overexpressing exosomes specification (my harvesting from cells exposed to the inhibitor in the presence of Ad-KLF2, for example)?
3. There is a lack of a key control. Cells in the absence of exosomes are needed for many of the graphs. i.e. do the control exosomes do anything already to the cells?
4. KLF-2 does many things. Are the authors sure that it is miRNA in the exosomes that mediate the effects. They do not directly prove this (comment 2 relevant).
5. The in vivo experiment surely needs the miRNAs pre-loading into synthetic exosomes? Or the experiment is simply to test the control and KLF-overexpressing exosomes by injection in vivo once purified.
6. An exosome expert should read over the purification and characterisation of the exosomes.
7. Validations of the microarray is lacking and the RNA profiling.
8. Informatics should be used to overlay targets for the 2 selected miRNAs and determine if the genes they overlay with are responsible for the effect
9. We see a little bit of uptake of the exosomes, but what are the dynamics? Does this actually happen in vivo? This is hard to test, but one could assess by injection of labelled exosomes and uptake into the lung EC.

NCOMMS-19-09015A Sindi et al.

Response to the Reviewers' comments

We are grateful to the Reviewers for their helpful comments.

Reviewer #1 (Remarks to the Author):

In this paper by Sindi et al, the authors show that flow-stimulated KLF2 produces exosomes that seem to be protective, and attribute this to the presence of miR-181 and miR-324 regulation of Notch and Ets.

The study is very well-performed and links this to the clinical situation by studying cells derived from IPAH patients. A few minor points:

1. The authors never show a direct interaction between the miRs delivered and the the Ets/Notch mRNA. Perhaps a luciferase assay will confirm that which miR is targeting which gene specifically.

Response: We have performed luciferase reporter assays to show direct interaction of miR-181/324 with 3'UTR of ETS-1 and Notch4 mRNA (please see Results page 9, Methods page 22, Supplementary Methods page 14 and Supplementary Figure S14 B). miRNA binding sites are shown in the Supplementary Figure S14A.

2. The transfer of exosomes is thought to mediate these effects, but could the authors show that it is purely the exosomes by inhibiting exosome transfer (GW compound)?

Response: Following the Reviewer's suggestion, we treated KLF2-overexpressing HPAECs with GW4869 to demonstrate that the inhibition of exosome transfer completely abrogated exosome-induced responses in HPASMCs (please see Results page 8, Figure 4d and Supplementary Methods page 16)

3. While the authors show that miRs are indeed transferred, are the specific miRs actually increased in the recipient cells? Numerous studies show that individual miR levels are quite low and do not necessarily transfer efficiently.

Response: We have shown that transfer of KLF2-induced exosomes increased intracellular levels of miR-181 and miR-324 in HPAECs (Figure 3 e, f) and HPASMCs (Figure 4 b, c). Moreover, exosomes obtained from HPAECs transfected with specific inhibitors of miR-181 and miR-324 had no effect on endothelial and smooth muscle cell responses (please see Supplementary Figure S10 and Figure 4).

4. The authors show a strong effect on GlcNAc signalling but this does not seem to be investigated further. Are there specific proteins that are affected post-translationally by interruption of GlcNAc signaling?

Response: miR-181 and miR-324 may alter post-translational modification of proteins via O-linked (on Ser/Thr) or N-linked attachment (on Asn) of GlcNAc or via attachment of GlcNAc onto fucose residues on the Notch receptors, essential for their activation and ligand specificity. While O-GlcNAcylation is required for vascular adaptation to stress conditions, a chronic response has been linked to vascular injury and increased vasoconstrictor/reduced vasodilator responses in cardiovascular diseases, including arterial hypertension. We have now included this in Discussion on pages 14 and 15.

In this study we show that specific cell responses result from miRNA-mediated decrease in ETS-1 and Notch4 expression but do not exclude a possibility that other pathways and targets may contribute to the beneficial effects of miR-181 and miR-324. Preliminary analysis of western blots (data not shown) shows that overall levels of O-GlcNAcylated proteins in miR-181 and miR-324-transfected cells are reduced but identification of specific targets and delineating their roles in KLF2 signaling is beyond the scope of this manuscript.

5. Many journals now require publication of full blots, at least in the supplement. The authors should comply with this growing trend.

Response: We have provided all blots in Supplemental Data Files.

Reviewer #2 (Remarks to the Author):

In this submitted manuscript, Sandi et al. profiled the exosomal miRs released from KLF2-overexpressed human pulmonary endothelial cells. In reference to the down-regulated miRs in human PAH and rodent PH models, the authors identified that miR-181a-5p and miR-324-5p are endothelium protector in the context of PAH. They also explored the endothelial-endothelial and endothelial-smooth muscle cells communications via KLF2 induced exosomes. Further, ETS-1 and NOTCH4 were suggested to be the downstream targets of miR-181 and 324, respectively. These key miRNA and molecules were studied in the human PAH lung specimens. MiR-181 and miR-324 supplements in the sugen+hypoxia PH mice model mitigated PH severity.

Overall, the study is nicely executed and the conclusion is convincing. Addressing the following points could further strengthen the manuscript.

1. The involvement of KLF2, miR-181, ETS-1 and Notch4 in PAH has been well documented. The authors need to discuss the novelty of the deduced molecular axis.

Response: The novelty of our approach is summarised in Discussion on page 13.

We are first to provide evidence that KLF2 signaling is a common feature of human PAH and describe therapeutic, cooperative actions of KLF2-induced endothelial miRNAs involving key regulators of vascular homeostasis, Notch4 and ETS1.

While Notch4 expression was shown to increase in hypoxic rats (Qiao L et al. PLoS One. 2012;7(12):e51514), its effects and its role in human PAH have not been documented. ETS1 has only been implicated indirectly from its role in angiogenesis, glycolysis and other processes relevant to PAH (and therefore mentioned in reviews i.e. Carman et al. Am J Pathology 2019 Volume 189, Issue 6, Pages 1133–1144), but we are first to show increased expression of ETS1 in PH animals, cells and tissues from PAH and delineate KLF2-miR181/miR324-Notch4 and ETS-1 signaling axis.

2. The authors showed that KLF2-associated exosomes mimics the effects of KLF2 overexpression and shear stress in ECs. I am wondering, whether the KLF2 increased in those exosome- treated ECs? In the recipient cells, is the effects of exosomes KLF4 dependent or independent? The authors

should verify whether the KLF2 mRNA or even KLF2 adenovirus vesicles were contained in those exosomes.

Response: We did not detect a significant increase in KLF2 expression in cells treated with KLF2 exosomes (please see new Supplementary Figure S2). While we found a small number of adenoviral particles in the concentrated exosomal fraction from adenovirus-infected cells (please see Results page 5, Supplemental Figure S2 and S3, Supplemental Methods page 4 and 5), their effect on KLF2 expression in the recipient cells was negligible.

KLF4 silencing in HPAECs did not significantly affect responses induced by KLF2 exosomes (please see Results page 6, Supplemental Figure S6, Supplemental Methods page 10), suggesting that the observed effects were KLF4-independent.

3. With KLF2 overexpression in EC, the amount of miR-181 and miR-324 is higher in exosomes than that in the donor cells. Also, the export ratio is higher than 1. These results indicate that KLF2 increases the secretion of these two miRs in exosomes. It is important to address the mechanism by which KLF2 enhances miR-181 and -324 expression and secretion.

Response: Potential mechanisms by which KLF2 may enhance miR-181 and miR-324 expression and secretion are now discussed on page 15:

“The mechanism through which KLF2 upregulates expression of miR-181 and miR-324 is unknown and may involve interaction of KLF2 with promoter or enhancer regions of target miRNAs, indirect transcriptional regulation by intermediate agents or CpG demethylation factors resulting in hypomethylation of pri-miRNA promoters or post transcriptional processes.”

Although the mechanism of miRNA sorting is not well understood, specific consensus sequence motifs (EXOmotifs) have been implicated. miR-181a-5p has two EXOmotifs that differ from consensus sequence in one nucleotide (GGAC, UGAC), while miR-324-5p shows three motifs, one with perfect match to consensus sequence (CCCU) and two showing alteration in one nucleotide (UCCG, GGAC) (please see Discussion on page 15). Delineating specific mechanisms responsible is beyond the scope of this investigation, but we hope to address this question in future studies.

4. KLFs may have distinct effect in ECs and SMCs. The authors need to determine whether the KLF2-associated exosomes from EC induces KLF2 in SMCs? What're the functional outcomes if they directly overexpress KLF2 in SMCs?

Response: we did not find KLF2 expression changes in HPASMCs treated with exosomes from KLF2-overexpressing HPAECs (please see Results page 8 and new Supplementary Figure S12).

In contrast to the effects induced by endothelial KLF2 exosomes, direct overexpression of KLF2 did not abrogate PDGF-induced HPASMC proliferation (Figure S12B, C in the Online Data Supplement), indicating that KLF2 effects are cell-type specific (Results page 8).

5. With in silico analysis predict that ETS-1 and NOTCH4 mRNA are direct targets of miR-181 and miR-324, biological validation of this prediction is absolutely necessary. Moreover, authors need to show whether these miRs and their targeting sequences are conserved between human and mouse.

Response: Changes in mRNA expression of selected targets were validated by qPCR (please see Results page 9 and Supplementary Figure S13). We have also performed luciferase reporter assays showing targeting of 3'UTR of ETS-1 and Notch4 by miR-181 and miR-324 (please see Results page 9, Methods page 22, Supplementary Methods page 14 and Supplementary Figure S14). miRNA binding sites are shown in the Supplementary Figure S14A.

The seed sequences of miR-181 and miR-324 families are conserved. In-silico analysis of putative binding for miR-181 (MIPF0000007) and miR-324 (MIPF0000165) on 3'UTR sequence of genes was performed using 13 independent programs (miRWalk, miRDB, PITA, MicroT4, miRMap, RNA22, miRanda, miRNAMap, RNAhybrid, imRBridge, PICTAR2 and Targetscan) with miRWalk v2.0 web tool. In-silico evidence from at-least 1 of the 13 programs identified conserved binding of miR-181 family members 181a,181b,181c and 181d to 3'UTR Notch4 sequence in both mouse and human. Binding of miR-324 to the 3'UTR sequence of ETS-1 was also conserved in mouse and human.

6. The critical molecules in their proposed pathway were verified in the human PAH lung specimens, which is excellent. To further strengthen these results, FISH experiment is suggested to detect miR-181 and miR-342 by in patients' specimens.

Response: Considering that miR-181 and miR-324 are ubiquitously expressed, optimization of the technique and then quantification of these miRNAs in human lung sections (of which we have a very limited supply) using FISH is likely to be inconclusive. However, we have included additional analysis showing decreased protein expression of KLF2 and increased expression of ETS-1 and Notch4 in human PAH lung tissues (please see Supplemental Methods page 2, Supplementary Figure S17).

7. Since the majority of works are carried out on exosomes, is it possible to deliver human ECs released KLF4 -associated exosomes to PH mice? It will be a more direct evidence to show the exosomal related therapeutic effect.

Response: Potential homeostatic effects of KLF4-induced exosomes have not been investigated and will be addressed in future studies. The complexity of exosomal cargo and low production yield are obstacles for clinical translation and therefore we focused our attention on identification and delivery of exosomal miRNAs. We now clarify this in Introduction on page 4 and in Discussion on page 3.

8. Since low flow is associated with PAH (ref. 20). Additional control in Fig. 3h-j should include low flow experiment. A physiological level of flow, e.g., 10 dyne/cm² should also be used in experiments in Fig. S3.

Response: We have included 10 dynes/cm² in the new Supplementary Figure S9. Both values, 4 and 10 dynes/cm² are within a physiological range for medium size pulmonary arteries (Tang et al. *Pulm Circ.* 2012;2:470-476) and stimulate KLF2 expression.

Modeling of flow in PAH lung is complex, with a decrease expected in the large-to medium vessels and an increase in the smallest, occluded distal vasculature. Here we argue for a potential therapeutic effect of KLF2 activation induced by physiological, laminar flow and therefore chose to work with human pulmonary artery endothelial cells and a physiological range of flow characteristic for this cell type. KLF2 activity also depends on a flow pattern (Wang et al. *Biochem*

Biophys Res Commun. 2006 Mar 24;341(4):1244-51) and therefore modelling flow conditions which affect KLF2 activation in PAH would require more extensive studies.

9. Heatmaps generated from RNA-seq should be included in Fig. 5. The protein levels of KLF2, ETS-1, and Notch4 in the lung tissues from PAH patients are necessary to be shown. The authors need to identify cell types shown in Figure 8j?

Response: Heatmaps are now shown in Figure 5c and protein levels of KLF2, ETS-1 and Notch 4 in PAH lung lysates are shown in Supplementary Figure S17.

In Figure 8j, arrows point to endothelial cells facing the vessel lumen. Endothelial cells were identified by co-localization of KLF2 with vWF. Additional images are provided in Data Supplement (S17).

10. KLF2 overexpression increases exosomal miR-181 and miR-324. However, the total exosome number were not changed. The authors need to use an additional control, namely, a KLF2-decreased miRNA (chosen from the heatmap in Fig. 2) to show the specificity of KLF2 in miR-181 and miR-324 biogenesis, but not exosome formation and secretion.

Response: Following the Reviewer's suggestion, we used miR-32-5p, which was significantly downregulated by KLF2 (Supplemental Table S1). PCR analysis showed an increase in the exosomal and intracellular levels of miR-181 and miR-324, while miR-32-5p levels were reduced in KLF2-overexpressing cells (please see Results page 8, Supplementary Figure S11 and Supplementary Table S1). Figure 2 shows KLF2-induced changes in miRNA expression in exosomes. We have realised that in the previous version of the manuscript the label below the heatmap in Figure 2 suggested that the analysis was done in KLF2-overexpressing cells, not in exosomes from KLF2-overexpressing cells. We have now corrected the figure labels to clarify this.

11. If authors secure KLF2 mutant ECFCs from PAH patients, then they should overexpress miR-181a-5p and miR-324-5p to determine whether the PH phenotype of KLF2 mutant ECFCs can be altered.

Response: While we were unable to obtain ECFCs from HPAH patients with KLF2 mutation, we overexpressed p.H288Y mutant in human pulmonary endothelial cells to demonstrate loss of KLF2 function caused by this mutation (please see Results Page 10, Discussion page 16, Supplementary Figure S18 and Supplemental Methods page 10-11)

Reviewer #3 (Remarks to the Author):

This manuscript addresses the mechanisms that might be relevant to KLF2 in PAH. This gene was recently associated. The authors assess KLF-2 using an adenovirus to overexpress the protein then purify exosomes from HPAEC and control cells and then perform a series of studies to look at function of the control and KLF2 overexpressing exosomes on PAEC function, functions that are relevant to PAH. They then assess miRNAs in those exosomes by microRNA array and discover 100 or so that are changed. After a series of steps then bring this down to 8, then 2 (181a-5p and 324-5p) and perform more studies in vitro and with miRNA supplementation in vivo to assess effects and

effects on a couple of key target genes. The manuscript has a number of novel angles, yet the following are relevant comments.

1. There is a disconnect between the beginning, the middle and then the end. It starts with KLF2 genetics (and then that is forgotten), has exosomes and the effect of KLF2 overexpressing exosomes (and then forgotten) and then miR supplementation in the absence of being excapsulated in exosomes. This is limitation to the story telling, as they become different questions. This needs to be addressed. For example, exosomes in the in vivo supplementation, of induce the KLF mutation seen in the genetic form and assess the relevance to the effect. Something linking all this is lost across the paper.

Response: We have added new text in Introduction on page 3-4 and in Discussion on page 13 that will hopefully help improve the storyline and highlight the novelty of our approach.

“Data from pre-clinical studies and discovery of a missense mutation in KLF2 gene in human HPAH suggest that KLF2 function may be compromised in PAH.” The main aim of this study was to identify a treatment strategy that would mimic homeostatic effects of endothelial KLF2 in the vasculature.

While viral delivery of KLF2 in PH rats (observations published only as a conference abstract) or treatment with KLF2-induced exosomes (only observed in a mouse model of atherosclerosis) can offset some of the effects of KLF2 inhibition, they have limited applicability in clinical practice. The complexity of exosomal cargo and low production yield are obstacles for clinical translation (Gangadaran P et al. Front Pharmacol. 2018;9:169). We therefore carried out an unbiased screen of KLF2-induced exosomal miRNAs with the view of establishing a more viable treatment strategy.

“We are first to provide evidence of dysregulated KLF2 signaling in human PAH and describe therapeutic, cooperative actions of KLF2-induced endothelial miRNAs involving key regulators of vascular homeostasis, Notch4 and ETS1.”

2. There is a distinct lack of inhibition studies. Why not take out the miRNA from KLF-2 overexpressing exosomes specification (my harvesting from cells exposed to the inhibitor in the presence of Ad-KLF2, for example)?

Response: Inhibitors of miR-181 and miR-324 completely prevented the effects of KLF2 exosomes in HPAECs (please see Results page 7-8 and Supplementary Figure S10). We also used an inhibitor of exosome formation, GW4869, which also inhibited exosome-induced responses in HPASMCs co-cultured with KLF2-overexpressing HPAECs (please see Results page 8 and Figure 4 and Supplemental Methods page 16).

3. There is a lack of a key control. Cells in the absence of exosomes are needed for many of the graphs. i.e. do the control exosomes do anything already to the cells?

Response: We did not observe significant responses between the untreated controls, exosome controls (cells treated with control exosomes) or transfection controls (please see Results page 6 and Supplementary Figure S5). In the interest of clarity, in most graphs we showed only treatment controls.

4. KLF-2 does many things. Are the authors sure that it is miRNA in the exosomes that mediate the effects. They do not directly prove this (comment 2 relevant).

Response: please see the response to the Reviewer's query in point 2.

5. The in vivo experiment surely needs the miRNAs pre-loading into synthetic exosomes? Or the experiment is simply to test the control and KLF-overexpressing exosomes by injection in vivo once purified.

Response: miRNAs were loaded into liposomal transfection reagent, InvivoFectamine, a widely used agent for high efficiency in vivo delivery of siRNA and miRNA.

6. An exosome expert should read over the purification and characterisation of the exosomes.

7. Validations of the microarray is lacking and the RNA profiling.

Response: Changes in the expression of selected miRNAs and their targets analysed by qPCR are shown in Supplementary Figure S11 and Supplementary Figure S13.

8. Informatics should be used to overlay targets for the 2 selected miRNAs and determine if the genes they overlay with are responsible for the effect

Response: Expression of haematopoietic cell early activation antigen CD69, identified as a common target of miR-181 and miR-324 (Table S3), was not detectable at protein level in HPAECs (data not shown). This is stated in Results on page 9.

9. We see a little bit of uptake of the exosomes, but what are the dynamics? Does this actually happen in vivo? This is hard to test, but one could assess by injection of labelled exosomes and uptake into the lung EC.

Response: We have labelled exosomes and show their distribution in different organs 4h post-injection (please see Supplemental Material and Methods page 7 and Supplementary Figure S7). Vascular localization of exosomes is marked by co-localization with vWF. Therapeutic exosome delivery has been shown in several pre-clinical models (Gareth R. Willis et al. Am J Respir Crit Care Med Vol 197, Iss 1, pp 104–116, Jan 1, 2018; Lydia Alvarez-Erviti et al. Nature Biotechnology 29, 341-345, 2011) and exosome-mediated transfer of small RNAs, resulting in changes in gene expression.

Rather than injecting exosomes, we aimed to identify exosomal miRNAs of potential therapeutic significance. We have previously shown endothelial localization of fluorescently-labelled siRNA by liposomal in vivo delivery (Abdul-Salam et al. Circ Res. 2019 Jan 4;124(1):52-65) and here we demonstrated that miR-181 and miR-324 levels in the lung increase following the in vivo transfection, while expression of their target genes is reduced.

REVIEWERS' COMMENTS:

Reviewer #1 (Remarks to the Author):

I think the authors were quite responsive to the original critiques and have added adequate new data.

Reviewer #2 (Remarks to the Author):

The authors have addressed my critics and it is ready to be accepted by Nature Communication.

Reviewer #3 (Remarks to the Author):

No further comments